# Gut microbiota and fecal short chain fatty acids differ with adiposity and country of origin: the METS-microbiome study

Gertrude Ecklu-Mensah [1,15], Candice Choo-Kang [2,15], Maria Gjerstad Maseng[3,4,5], Sonya Donato [1], Pascal Bovet [6,7], Bharathi Viswanathan[7], Kweku Bedu-Addo[8], Jacob Plange-Rhule[8], Prince Oti Boateng[8], Terrence E. Forrester [9], Marie Williams[9], Estelle V. Lambert[10], Dale Rae[10], Nandipha Sinyanya[10], Amy Luke [2], Brian T. Layden[11,12], Stephen O'Keefe[13], Jack A. Gilbert [1] ✉ & Lara R. Dugas [2,14] ✉

The relationship between microbiota, short chain fatty acids (SCFAs), and obesity remains enigmatic. We employ amplicon sequencing and targeted metabolomics in a large ($n$ = 1904) African origin cohort from Ghana, South Africa, Jamaica, Seychelles, and the US. Microbiota diversity and fecal SCFAs are greatest in Ghanaians, and lowest in Americans, representing each end of the urbanization spectrum. Obesity is significantly associated with a reduction in SCFA concentration, microbial diversity, and SCFA synthesizing bacteria, with country of origin being the strongest explanatory factor. Diabetes, glucose state, hypertension, obesity, and sex can be accurately predicted from the global microbiota, but when analyzed at the level of country, predictive accuracy is only universally maintained for sex. Diabetes, glucose, and hypertension are only predictive in certain low-income countries. Our findings suggest that adiposity-related microbiota differences differ between low-to-middle-income compared to high-income countries. Further investigation is needed to determine the factors driving this association.

Obesity remains an ongoing global health epidemic that continues to worsen, affecting more than 600 million adults worldwide[1], including over a third of Americans[2]. Importantly, comorbidities associated with obesity account for over 60% of deaths worldwide[3]. A major driver of obesity is the adoption of a Western lifestyle, which is characterized by excessive consumption of ultra-processed foods[4–6]. Obesity has been accompanied by dramatic increases in the prevalence of non-communicable diseases such as type two diabetes and hypertension among people of African origin[5–9]. Therefore, disrupting the rapidly expanding obesity epidemic, particularly among African-origin

[1]Department of Pediatrics, Center for Microbiome Innovation, University of California San Diego, La Jolla, CA, USA. [2]Public Health Sciences, Parkinson School of Health Sciences and Public Health, Loyola University Chicago, Maywood, IL, USA. [3]Institute of Clinical Medicine, Faculty of Medicine, University of Oslo, Oslo, Norway. [4]Dep. of Gastroenterology, Oslo University Hospital, Oslo, Norway. [5]Bio-Me, Oslo, Norway. [6]University Center for Primary Care and Public Health (Unisanté), Lausanne University Hospital, Lausanne, Switzerland. [7]Ministry of Health, Victoria, Republic of Seychelles. [8]Department of Physiology, SMS, Kwame Nkrumah University of Science and Technology, Kumasi, Ghana. [9]Solutions for Developing Countries, University of the West Indies, Mona, Kingston, Jamaica. [10]Research Unit for Exercise Science and Sports Medicine, University of Cape Town, Cape Town, South Africa. [11]Department of Medicine, University of Illinois at Chicago, Chicago, IL, USA. [12]Jesse Brown Veterans Affairs Medical Center, Chicago, IL, USA. [13]Department of Medicine, University of Pittsburgh, Pittsburgh, PA, USA. [14]Division of Epidemiology and Biostatistics, School of Public Health, Faculty of Health Sciences, University of Cape Town, Cape Town, South Africa. [15]These authors contributed equally: Gertrude Ecklu-Mensah, Candice Choo-Kang. ✉e-mail: jagilbert@health.ucsd.edu; ldugas@luc.edu

populations, is critical to controlling the cardiometabolic disorder epidemic[10]. However, successfully managing and treating obesity and its comorbidities, and specifically maintaining weight loss long-term, is particularly challenging due to an incomplete understanding of the heterogeneous and complex etiopathology, as well as additional challenges facing populations experiencing rapid urbanization[10–12]. The epidemiologic transition is a model able to capture these shifts in dietary and rural-to-urban movements and is characterized by diets that are high in ultra-processed foods with a significant loss in fiber, as evidenced in the United States, where less than 50% of the population meet dietary fiber recommendations[13].

Gut microbial ecology and metabolism play pivotal roles in the onset and progression of obesity and its related metabolic disorders[14]. Obese and lean individuals have reported differences in the composition and functional potential of the gut microbiome, with an overall reduction in species diversity in the obese gut[7,15–19], additionally, fecal microbiota transfer from obese donors to mouse models can recapitulate the obese phenotype[20–22]. Further, fecal microbiota transplant from healthy donors into patients with obesity and metabolic syndrome has been shown to improve markers of metabolic health in the recipients[23]. While these studies suggest that modification of microbial ecology may offer new options for the treatment and prevention of obesity, the mechanism that drives the microbiota-obesity relationship is not fully understood. The microbiota may facilitate greater energy exploitation from food and storage capacity by the host[20,24], influencing adipose tissue composition and fat mass gain, as well as providing chronic low-grade inflammation and insulin resistance[25,26].

Among the numerous microbial metabolites modulating obesity, there is an ever-growing interest in the role of short-chain fatty acids (SCFAs), which includes butyrate, acetate, and propionate as potential biomarkers for metabolic health as well as therapeutic targets. SCFAs derive primarily from microbial fermentation of non-digestible dietary fiber in the colon. They have many effects on host metabolism, including serving as an energy source for host colonocytes, used as precursors for the biosynthesis of cholesterol, lipids, and proteins, and regulating gut barrier activities[27–29]. Human and animal studies demonstrate a protective role of SCFAs in obesity and metabolic disease. In experimental animal models, SCFA supplementation reduces body weight, improves insulin sensitivity, and reduces obesity-associated inflammation[30–34]. In humans, increased gut production of butyrate correlates with improved insulin response after an oral glucose tolerance test[35]. Although increased SCFA levels are generally observed as positive for health[36], other studies have suggested that overproduction may promote obesity, possibly resulting from greater energy accumulation[37–41]. Indeed, a previous study observed greater fecal SCFA concentrations to be linked with obesity, increased gut permeability, metabolic dysregulation, and hypertension in a human cohort[42].

The conflicting obesity role of SCFAs identified by existing studies may result from the variation in the gut microbiota, which is shaped by lifestyle and diet. Adequately powered studies in well-characterized populations may permit more rigorous assessments of individual differences. Prior comparative epidemiological studies have broadly focused on either contrasting the gut microbiota of extremely different populations, such as the traditional hunter-gatherers and urban-westernized countries, or ethnically homogenous populations[43–46]. Demographic factors represent one of the largest contributors to the individualized nature of the gut microbiome[46–48]. The five diverse, well-characterized cohorts from the modeling of the epidemiologic transition study (METS) offer a unique opportunity to examine the issues since they are more representative of most of the world's population. METS has longitudinally followed an international cohort of approximately 2500 African origin adults spanning the epidemiologic transition from Ghana, South Africa, Jamaica, Seychelles, and the US since 2010 to investigate differences in health outcomes utilizing the framework of the epidemiologic transition. Pioneering microbiome studies from the METS

cohorts reveal that cardiometabolic risk factors, including obesity, are significantly associated with reduced microbial diversity and the enrichment of specific taxa and predicted functional traits in a geographic-specific manner[7,49]. While yielding valuable descriptions of the connections between the gut microbiota ecology and disease, particularly obesity, as well as pioneering the efforts of microbiome studies of populations of African origin on different stages of the ongoing nutritional epidemiologic transitions, these studies, however, have applied small sample size ($N = 100$ to $N = 655$), and also did not utilize all the countries in the METS cohort. Thus, uncertainties remain as to the precise interpretation of the microbiome-obesity associations, which hampers further progress toward diagnostic and clinical applications.

Our new study, METS-Microbiome, investigated associations between the gut microbiota composition and functional patterns, concentrations of fecal SCFAs, and obesity in a large ($N = 1904$) adult cohort of African origin, comprised of Ghana, South Africa, Jamaica, Seychelles, and the US spanning the epidemiologic transition[50,51]. The central hypothesis is that alterations to the gut microbiota and community composition will be associated with increasing stages of the epidemiologic transition, reductions in fecal SCFA levels, and higher obesity prevalence. Here, we show profound variations in gut microbiota, including significant changes in community composition, structure, and predicted functional pathways as a function of population obesity and geography. Importantly, the utility of the microbiota in predicting whether an individual is non-obese or obese differs considerably by country of origin and suggests that lifestyle traits in high-income countries may increase obesity risk even for lean individuals. Overall, our findings are important for understanding the complex relationships between the gut microbiota, population lifestyle, and the development of obesity, which may set the stage for defining the mechanisms through which the microbiome may shape health outcomes in populations of African origin.

## Results
### Obesity differs significantly across the epidemiological transition
From 2018 to 2019, the METS-Microbiome study recruited 2085 participants (~60% women) ages 35-55 years old from five different sites (Ghana, South Africa, Jamaica, Seychelles, and USA). Of these participants, 1249 have been followed on a yearly basis since 2010 under the parent METS study. Data from 1,867 participants with complete data sets were used in this analysis. The overall mean age was $42.5 \pm 8.0$ years (Table 1). Mean fasted blood glucose was $105.2 \pm 39.4$ mg/dL, mean systolic blood pressure was $123.4 \pm 18.1$ mm Hg, and mean diastolic blood pressure was $77.2 \pm 13.1$ mm Hg. When compared to high-income countries (Jamaica, Seychelles and USA), women and men from low- and middle-income (Ghana and South Africa) had significantly lower BMI (except South African women), fasted blood glucose, and blood pressure (systolic and diastolic). Mean BMI was lowest in South African men ($22.3$ kg/m$^2 \pm 4.1$) and highest in US women ($36.3$ kg/m$^2 \pm 8.8$). When compared to the US, all sites had a significantly lower prevalence of obesity ($p < 0.001$ for all sites except for Seychelles: $p = 0.02$). The prevalence of hypertension was lowest in Ghanaian men (33.1%) and highest in US men (72.7%). The prevalence of diabetes was lowest in South African women and men (3.5% for women and men) and highest in Seychellois men (22.8%). When compared to the US, the prevalence of hypertension and diabetes was significantly lower in countries at the lower end of the spectrum of HDI (i.e., Ghana and South Africa) when compared to the US ($p < 0.001$).

### Microbial community composition and predicted metabolic potential differs significantly between countries and correlates with obesity
Following the removal of samples that had fewer than 6000 reads and featured less than ten reads in the entire dataset, a total of 433,364,873

**Table 1 | METS-microbiome participant's characteristics**

|  | Ghana | South Africa | Jamaica | Seychelles | US |
|---|---|---|---|---|---|
| *Women* | | | | | |
|  | $n = 254$ | $n = 228$ | $n = 263$ | $n = 196$ | $n = 213$ |
| Age (years) | 40.74 ± 8.1 | 35.56 ± 7.8 | 45.16 ± 7.5 | 43.84 ± 6.1 | 45.44 ± 6.4 |
| BMI (kg/m²) | 28.30 ± 5.9 | 33.42 ± 8.6 | 32.12 ± 7.3 | 30.32 ± 7.2 | 36.34 ± 8.8 |
| Obese (%) | 45.0% | 61.0% | 60.4% | 49.5% | 74.7% |
| SBP (mm Hg) | 117.1 ± 18.5 | 115.20 ± 17.1 | 126.08 ± 19.0 | 123.28 ± 17.8 | 124.19 ± 18.4 |
| DBP (mm Hg) | 70.53 ± 12.2 | 75.20 ± 12.1 | 79.41 ± 12.6 | 79.37 ± 14.4 | 81.52 ± 12.1 |
| Hypertensive (%) | 37.5% | 37.3% | 57.4% | 55.5% | 65.4% |
| Glucose (mg/dL) | 110.45 ± 62.7 | 89.17 ± 20.0 | 107.46 ± 39.1 | 111.35 ± 27.2 | 107.07 ± 44.0 |
| Diabetic (%) | 6.8% | 3.5% | 12.9% | 13.9% | 19.9% |
| *Men* | | | | | |
|  | $n = 117$ | $n = 171$ | $n = 133$ | $n = 164$ | $n = 107$ |
| Age (years) | 43.92 ± 8.7 | 36.53 ± 7.2 | 44.42 ± 7.5 | 44.57 ± 5.1 | 47.12 ± 5.5 |
| BMI (kg/m²) | 23.7 ± 4.4 | 22.26 ± 4.1 | 24.8 ± 5.3 | 28.46 ± 5.5 | 30.37 ± 8.2 |
| Obese (%) | 13.4% | 5.3% | 15.7% | 39.2% | 44.4% |
| SBP (mm Hg) | 121.28 ± 15.4 | 122.71 ± 15.5 | 129.23 ± 17.1 | 130.43 ± 16.2 | 130.67 ± 16.0 |
| DBP (mm Hg) | 68.02 ± 13.0 | 75.32 ± 11.1 | 78.07 ± 11.5 | 81.64 ± 12.1 | 82.37 ± 12.2 |
| Hypertensive (%) | 33.1% | 45.0% | 50.3% | 65.9% | 72.7% |
| Glucose (mg/dL) | 100.52 ± 19.4 | 94 ± 23.4 | 99.04 ± 33.1 | 124.26 ± 44.2 | 107 ± 36.2 |
| Diabetic (%) | 1.0% | 3.5% | 4.8% | 22.8% | 17.5% |

Data are presented as mean ± standard deviation for continuous variables and percentages (%) for categorical variables.

16 S rRNA gene V4 region sequences were generated from the 1873 fecal samples, which were clustered into 13,254 amplicon sequence variants (ASVs). Country of origin describes most of the variation in microbial diversity and composition, with significant differences in both alpha and beta diversity. Although there were major variations in alpha diversity between countries and a large degree of inter-individual variation within countries, Ghana showed significantly greater diversity for all the alpha diversity metrics (Observed ASVs, Shannon Diversity, and Faith's phylogenetic diversity) when compared to all other countries (Fig. 1a). Seychelles and US had the lowest alpha diversity (Fig. 1a; Kruskal–Wallis; FDR-corrected, $q < 0.05$). The stool microbiota alpha diversity of non-obese individuals was significantly greater when compared with that of obese individuals (Fig. 1b, Wilcoxon rank-sum; $p < 0.05$). Beta diversity was also significantly different between countries (Fig. 1c–f, Supplementary Table 2; weighted UniFrac distance; PERMANOVA; $R^2 = 0.118$; $p < 0.001$; unweighted UniFrac distance; PERMANOVA; $R^2 = 0.083$; $p < 0.001$) and obese group (weighted UniFrac distance; PERMANOVA; $R^2 = 0.001$; $p = 0.031$; unweighted UniFrac distance; PERMANOVA, $R^2 = 0.003$; $p < 0.001$).

Next, we compared fecal microbiota diversity between obese individuals with their non-obese counterparts within each country independently. Greater alpha diversity was detected in non-obese subjects in the Ghanaian (Observed ASVs, Faith PD) and South African cohorts (Observed ASVs) only (Supplementary Table 1; Wilcoxon rank-sum; $p < 0.05$). Similarly, significant differences in beta diversity between obese and non-obese microbiota were observed in Ghana (Unweighted UniFrac; PERMANOVA; $R^2 = 0.004$; $p < 0.05$), South Africa (Unweighted UniFrac; PERMANOVA; $R^2 = 0.007$; $p < 0.05$) and US (Weighted UniFrac; PERMANOVA; $R^2 = 0.007$; $p < 0.05$) data sets (Supplementary Table 2). These results suggest that the beta diversity differences observed in the Ghanaian and South African participants may partly be due to the differences in rare taxa, whereas, among the US participants, the differences may be related to differences in proportional dominant microbial taxa. Collectively, these observations suggest that the country is a major driver of the variance in gut microbiota diversity and composition among participants with or without obesity, with marked contributions from Ghana and South Africa and modest contributions from the US in the overall cohort.

We also examined whether the country of origin or obesity relates to the presence of specific microbial genera frequently used to stratify humans into enterotypes[52]. As expected, large differences in enterotype between the countries were observed. The *Prevotella* enterotype (P-type) was enriched on the African continent, with 81% and 62% in Ghanaians and South Africans, respectively, while *Bacteroides* enterotype (B-type) was dominant in the US (75%), Jamaican cohorts (68%), and comparable proportions of both enterotypes among individuals from Seychelles (Supplementary Table 3). Further, obese individuals displayed a greater abundance of B-type, whereas a higher proportion of the P-type was associated with the non-obese group (Supplementary Table 3). Consistent with this observation, the abundance of B-type correlated with higher BMI ($q = 0.004$) than P-type. Significantly greater diversity and increased levels of total SCFA were observed in participants in the P-type (Supplementary Table 3). The relative abundance of shared and unique features between the different countries illustrated by the Venn diagram showed that Ghana has the largest number of unique genera, and the US has the lowest (Fig. 1g).

**Microbial taxa differ significantly between countries and between lean and obese individuals**

In comparison with the US, South African fecal microbiota had a significantly greater proportion of *Clostridium*, *Olsenella*, Bacilli, and *Mogibacterium*; Jamaican samples had a significantly greater proportion of Bacilli, *Bacteroides*, Clostridia, *Dialister*, *Enterobacteriaceae*, and *Oscillospiraceae*; Seychelles samples had a significantly greater proportion of *Clostridium*, *Olsenella*, and *Haemophilus*; and Ghanaian samples had a significantly greater proportion of *Clostridium*, *Prevotella*, *Weisella*, Enterobacteriaceae and Butyricicoccaceae. The US samples had a significantly greater proportion of *Aldercreutzia*, *Anaerostipes*, *Clostridium*, *Eggerthella*, *Eisenbergiella*, *Ruminococcaceae*, and *Sellimonas* compared to the four countries (Fig. 2a and Supplementary Fig. 1; ANCOM-BC; log fold change (LFC) > 1.4; $q < 0.05$).

When adjusted for country, age, and sex (ANCOM-BC; $q < 0.05$), 38 ASVs were significantly different between obese and non-obese

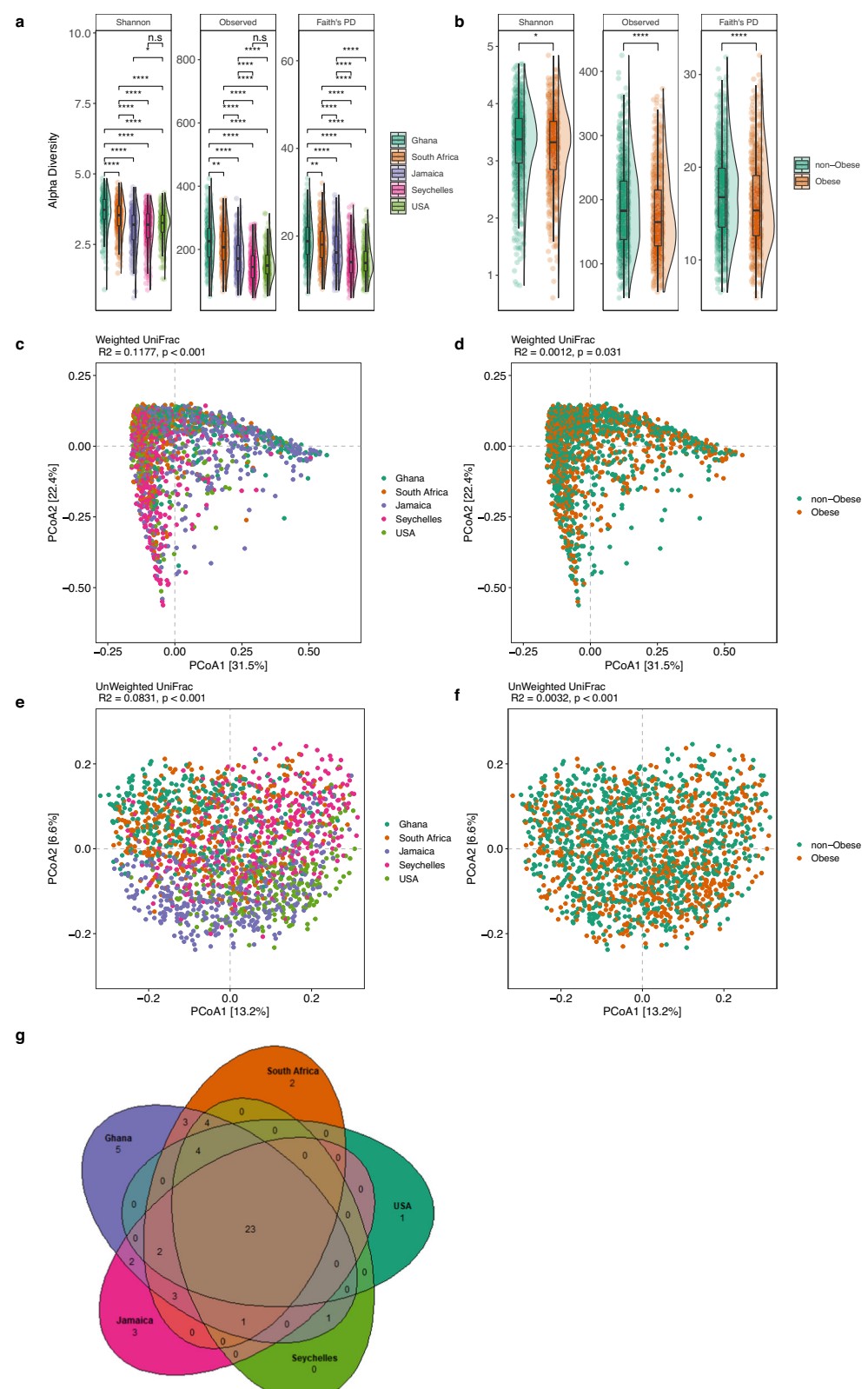

groups. The obese group was characterized by an increased proportion of *Allisonella*, *Dialister*, *Oribacterium*, *Mitsuokella*, and *Lachnospira*, whereas non-obese microbiota had a significantly greater proportion of *Alistipes*, *Bacteroides*, *Clostridium*, *Parabacteroides*, *Christensenella*, *Oscillospira*, *Ruminococcaceae* (UBA1819), and Oscillospiraceae (UCG010) (Fig. 2b).

Overall, there was almost no overlap in the features discriminating obese from non-obese groups between the country-specific

differential abundance analyses, except for a single *Parabacteroides* ASV that was differentially enriched in non-obese participants in both the Ghanaian and Jamaican cohorts (Fig. 2c, e; ANCOM-BC; *q* < 0.05). When comparing obese and non-obese groups in Ghana, 4 features, *Colidextribacter*, *Butyricicoccaceae, Oscillospiraceae*, and *Parabacteroides* were enriched in the non-obese group (Fig. 2c; ANCOM-BC; *q* < 0.05). The gut microbiota of the obese group in the South African cohort was enriched for 7 ASVs, including *Lactobacillus*, *Oribacterium*,

**Fig. 1 | Variation in gut microbiome diversity. a** Alpha diversity estimated by Shannon, Observed ASVs and Faith's PD (phylogenetic diversity) between countries. Exact false discovery rate (FDR)-corrected $q$ values from left to right: Shannon: 2.33e − 08, 3.55e − 41, 2.42e − 37, 1.25e − 31, 6.70e − 20, 1.03e − 16, 3.02e − 12, 0.015, 0.1089; Observed: 0.0022, 2.84e − 26, 4.77e − 51, 7.90e − 43, 7.60e − 16, 2.06e − 40, 1.61e − 32, 3.39e − 06, 5.79e − 06, 0.058; PD: 0.067, 1.51e − 11, 3.69e − 41, 3.30e − 41, 2.59e − 05, 2.04e − 30, 5.77e − 31, 3.33e − 14, 7.86e − 15. **b** Alpha diversity estimated by Shannon, Observed ASVs and Faith's PD (Phylogenetic Diversity) between obese and non-obese. Exact FDR-corrected $q$ values from left to right: Shannon: 0.014; Observed: 1.86e − 05; PD: 6.05e − 06. Alpha diversity metrics (Faith's PD, Observed ASVs, and Shannon) are shown on the $y$-axis in different panels, while country or obese groups are shown on the $x$-axis. **c** Beta diversity principal coordinate analysis based on weighted UniFrac distance between countries. **d** Beta diversity principal coordinate analysis based on weighted UniFrac distance between obese and non-obese. **e** Beta diversity principal coordinate analysis based on unweighted UniFrac distance between countries. **f** Beta diversity principal coordinate analysis based on unweighted UniFrac distance between obese and non-obese. The proportion of variance explained by each principal coordinate axis is denoted in the corresponding axis label. **g** Venn diagram of shared and unique genera between the five countries detected at a relative abundance >0.001 in more than 50% of the samples. Box plots show the interquartile range (IQR), the horizontal lines show the median values and the whiskers extend from the hinge no further than 1.5*IQR. Each colored dot denotes a sample. Statistical significance adjusted for multiple comparisons using false discovery rate (FDR) correction is indicated: *, $P < 0.05$; **, $p < 0.01$; ***, $p < 0.001$; ****, $p < 0.0001$, ns, non-significant; across countries and obese groups (Kruskal–Wallis test and pairwise Wilcoxon rank sum test; two-sided) for alpha diversity or by permutational multivariate analysis of variance (PERMANOVA) for beta diversity. Source data are provided as a Source Data file. Alpha diversity analysis for country, $n = 1873$ samples (Ghana, $n = 373$; South Africa, $n = 390$; Jamaica, $n = 401$; Seychelles, $n = 396$; USA, $n = 313$) and obesity status, $n = 1764$ samples. For Beta diversity analysis, $n = 1764$ samples.

and *Megasphaera* (Fig. 2d; ANCOM-BC; $q < 0.05$). In the Jamaican non-obese group, 6 ASVs were enriched including Christensenellaceae, *Desulfovibrio*, *Eubacterium* and *Parabacteroides*, whereas the relative proportion of *Ruminococcus* was greater in the obese Jamaican group (Fig. 2e; ANCOM-BC; $q < 0.05$). The US non-obese group was enriched for *Intestinimonas* and *Ruminiclostridium* when compared with their obese counterparts (Fig. 2f; ANCOM-BC; $q < 0.05$), whereas there were no significantly enriched features that discriminated obese from the non-obese group in participants from Seychelles.

## Microbial taxonomic features predict obesity overall but not within each country

Using supervised Random Forest machine learning, the predictive capacity of the gut microbial features to stratify individuals to country of origin, sex, or metabolic phenotypes was assessed. The predictive performance of the model was calculated by area under the receiver operating characteristic curve (AUC) analysis, which showed a high accuracy for country of origin (AUC = 0.97) (Fig. 3a and Supplementary Table 4), and a comparatively lower level of predictive accuracy for the obese state (AUC = 0.65) (Fig. 3b and Supplementary Table 5). Diabetes status was predicted with AUC = 0.63 (Fig. 3c and Supplementary Table 4), glucose status with AUC = 0.66 (Fig. 3d and Supplementary Table 5), hypertensive status with AUC = 0.65 (Fig. 3e and Supplementary Table 5) and sex with AUC = 0.75 (Fig. 3f and Supplementary Table 5). Random Forest analysis was also used to identify the top 30 microbial taxonomic features that differentiate between countries and obese states. Similar to the ANCOM-BC results, *Prevotella* and *Streptococcus* were at a greater proportion in the microbiota of Ghanaian and non-obese individuals, whereas *Mogibacterium* was at a greater proportion in the South African cohort (Supplementary Fig. 2a, b). A greater proportion of *Megasphaera* was associated with the Jamaican cohort, while a greater proportion of *Ruminococcaceae* was observed in the American microbiota (Supplementary Fig. 2a). *Weisella*, which was identified as having a significantly greater proportion in the Ghanaian cohort using ANCOM-BC, was observed to be a discriminatory feature for Seychelles microbiota using Random Forest (Supplementary Fig. 2a).

Similarly, the predictive capacity of gut microbiota features in stratifying individuals by sex or metabolic phenotypes was assessed separately for each of the five study sites. The predictive performance of the model calculated by AUC analysis showed changes in accuracy for all parameters determined for all sites (Fig. 3g–k and Supplementary Tables 6–11). For example, the obese state was marginally predictive only for Ghana (AUC = 0.57), while all other countries lost accuracy (Fig. 3g–k). The predictive accuracy (Supplementary Table 6) for diabetes status was only retained for Ghana (AUC = 0.69), and Jamaica (AUC = 0.66); glucose status prediction was lost for all countries but South Africa, where it improved (AUC 0.78); and prediction of hypertension was only retained for Ghana (AUC = 0.63). The predictive ability for sex was maintained for all countries (Supplementary Table 6).

## Predicted genetic metabolic potential differs by country and obesity status

The predicted potential microbial functional traits resulting from the compositional differences in microbial taxa between countries and obese states were assessed, although we acknowledge that currently available reference genome databases are likely biased toward well-studied Western populations and may have limited capacity to sufficiently characterize the gut microbiome from understudied populations[43,53]. Nonetheless, PICRUSt2 predicted a total of 372 MetaCyc functional pathways. ANCOM-BC analysis adjusted for sex, age, and BMI identified 67 pathways that accounted for discriminative features between the 4 different countries with the US (Supplementary Fig. 3a; ANCOM-BC; LFC > 1.4; $q < 0.05$). In comparison with the United States, MetaCyc pathways differentially increased in Ghana and Jamaica include methylgallate degradation, norspermidine biosynthesis (PWY-6562), gallate degradation I pathway, gallate degradation II pathway, histamine degradation (PWY-6185), toluene degradation III (via p-cresol) (PWY-5181). South African samples had a greater proportion of ʟ-glutamate degradation VIII (to propanoate) (PWY-5088), isopropanol biosynthesis (PWY-6876), creatinine degradation (PWY-4722), adenosylcobalamin biosynthesis (anaerobic) (PWY-5507), respiration I (cytochrome c) (PWY-3781) (Supplementary Fig. 3a; ANCOM-BC; $q < 0.05$). MetaCyc pathways linked to norspermidine biosynthesis (PWy-6562), mycothiol biosynthesis (PWY1G-0), were at a greater proportion in the Seychelles samples, whereas reductive acetyl coenzyme A (CODH-PWY), and chorismate biosynthesis II (PWy-6165) were depleted in the US samples (Supplementary Fig. 3a, ANCOM-BC; $q < 0.05$). ANCOM-BC analysis adjusted for site, sex, and age identified 24 predicted pathways that differentiated between obese and non-obese individuals (Supplementary Fig. 3b; ANCOM-BC; $q < 0.05$). Notably, the microbiota of non-obese individuals had a greater proportion of predicted pathways, including the TCA cycle, amino acid metabolism (P162-PWY, PWY-5154, PWY-5345), ubiquinol biosynthesis-related pathways (PWY-5855, PWY-5856, PWY-5857, PWY-6708, UBISYN-PWY), cell structure biosynthesis and nucleic acid processing (PWY0 845, PYRIDOXSYN-PWY) (Supplementary Fig. 3b; ANCOM-BC; $q < 0.05$). On the contrary, when stratified by country, no statistically significant predicted functional pathways differentiated non-obese from obese participants within each country except in Jamaica, where only a single predicted pathway (PWY7377) involved in adenosylcobalamin biosynthesis (anaerobic) was differentially enriched in non-obese individuals (Supplementary Fig 3c, ANCOM-BC; $q < 0.05$).

Next, KEGG orthology (KO) involved in pathways related to butanoate (butyrate) metabolism and LPS biosynthesis were

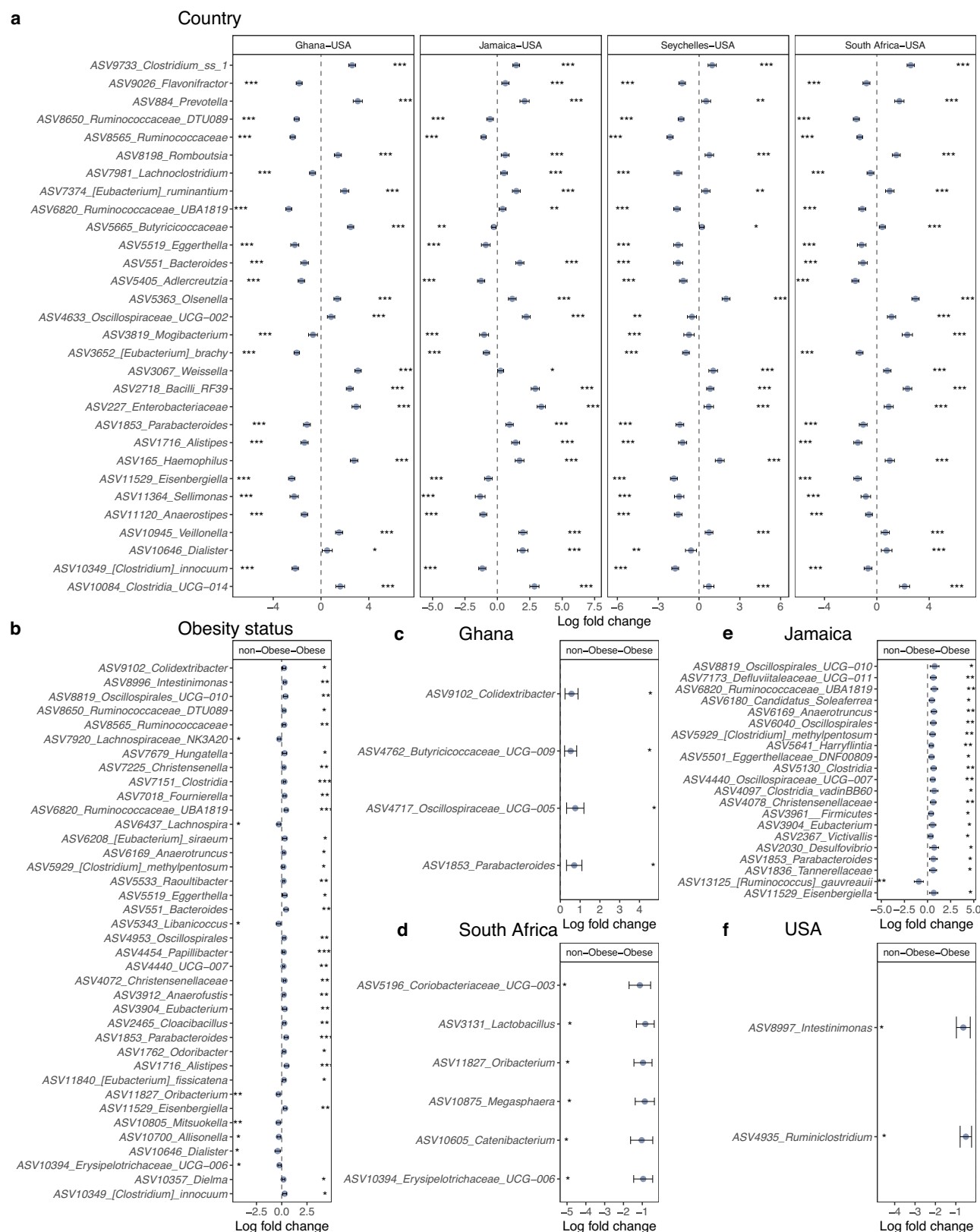

**Fig. 2 | Variation in gut microbiome composition. a** Differentially abundant taxa among countries with the US as the reference due to its status as the country with the highest HDI and highest obesity incidence ($n$ = 1694). **b** Differentially abundant taxa between obese and non-obese groups in the entire cohort ($n$ = 1694). Differentially abundant taxa between obese and non-obese groups within **c** Ghana ($n$ = 329); **d** South Africa ($n$ = 374); panel **e** Jamaica ($n$ = 386); panel **f** US ($n$ = 304). ANCOM-BC analyses adjusted for BMI, age, sex, and country. Data are presented by effect size (log fold change) with a 95% confidence interval (CI) calculated from the beta coefficient and standard errors estimated from the ANCOM-BC log-linear (natural log) model (two-sided; FDR-adjusted). The colored dot indicates effect size (log fold change), and the whiskers indicate 95% CI. Representative ASVs with log fold change >1.4 in at least one group are shown for the country. FDR-adjusted ($q$ < 0.05) effect sizes are indicated by * $q$ < 0.05; ** $q$ < 0.01; *** $q$ < 0.001. The exact $p$-values are available in the source data file. Source data are provided as a Source Data file. FDR false discovery rate, HDI human development index, ANCOM-BC analysis of compositions of microbiomes with bias correction.

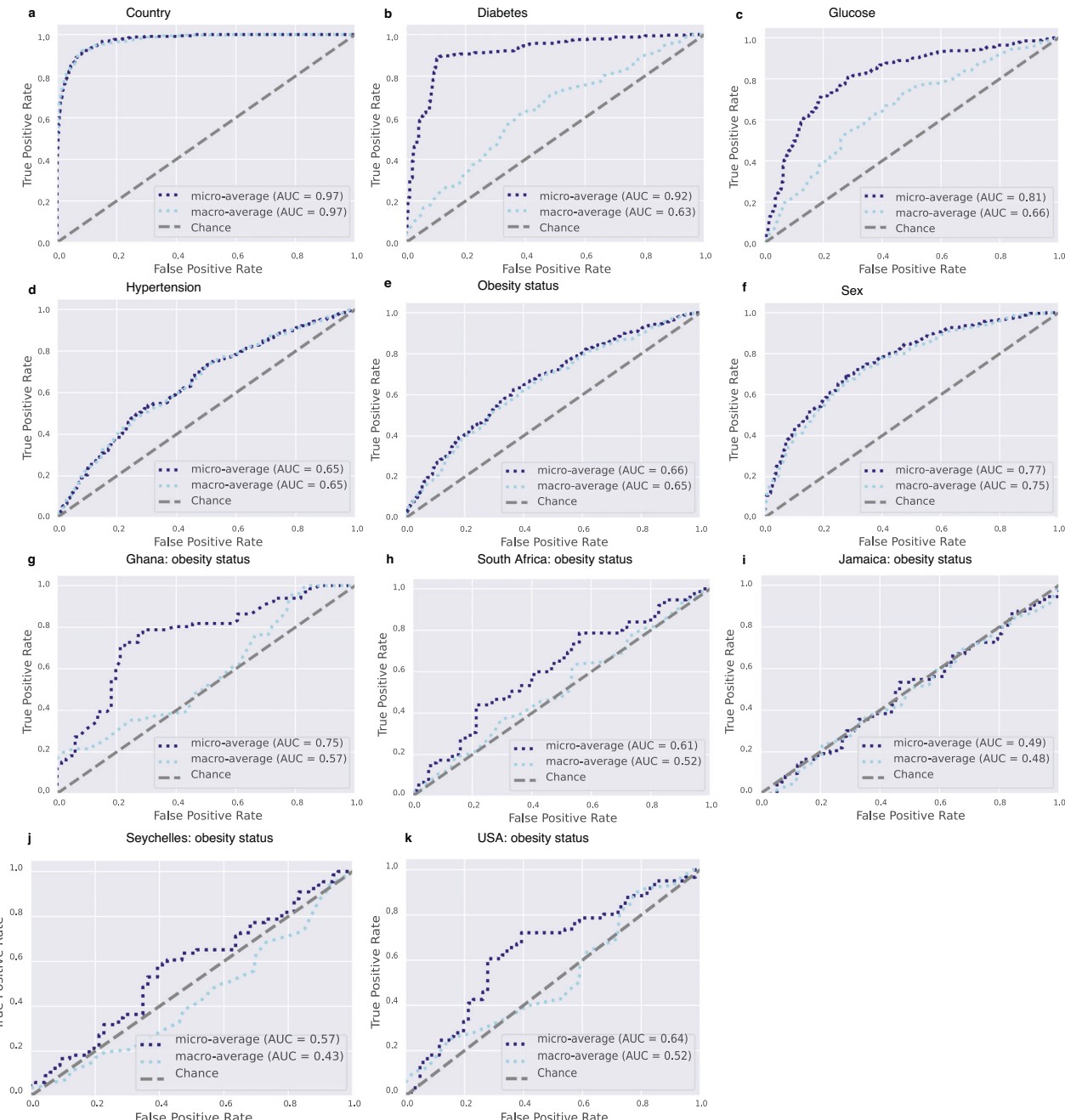

**Fig. 3 | Receiver operating characteristic curves showing the classification accuracy of gut microbiota in a Random Forest model.** Classification accuracy for estimating **a** all countries ($n = 1694$); **b** diabetes status ($n = 1657$); **c** glucose status ($n = 1657$); **d** hypertensive status (1694); **e** obesity status ($n = 1694$); **f** sex ($n = 1694$) are presented. Classification accuracy for estimating country-level obesity status in **g** Ghana ($n = 329$); **h** South Africa ($n = 374$); **i** Jamaica ($n = 386$); **j** Seychelles ($n = 361$); **k** USA ($n = 304$) are presented. Micro-averaging values are impacted by data imbalance since it averages across each sample, whereas Macro-averaging provides equal weight to the characterization of each sample. Macro-averaging values are reported in the text. AUC area under the curve.

investigated. Predicted genes involved in butyrate biosynthesis pathways showed that enoyl-CoA hydratase enzymes (K01825, K01782, K01692), lysine, glutarate/succinate enzymes (K07250, K00135, K00247), glutarate/Acetyl CoA enzymes (K00175, K00174, K00242, K00241 K01040, K01039) were differentially abundant in participants from Ghana, South Africa, Jamaica, and Seychelles in comparison to the US cohort (Supplementary Fig. 4a; ANCOM-BC; $q < 0.05$). The relative abundance of succinic semialdehyde reductase (K18121) was significantly increased only in South Africa, Jamaica, and the Seychelles population. Further, predicted genes proportionally abundant only in specific countries were observed. For instance, succinate semi-aldehyde dehydrogenase (K18119) was enriched only in the Ghanaian

cohort, 4-hydroxybutyrate CoA-transferase (K18122) was enriched among South African participants, and lysine/glutarate/succinate enzyme (K14268) differentially abundant within the Seychelles population (Supplementary Fig. 4a; ANCOM-BC; $q < 0.05$). The relative abundance of predicted genes encoding for enzymes such as maleate isomerase (K01799), 3-oxoacid CoA-transferase (K01027), and pyruvate/acetyl CoA (K00171, K00172, K00169) was greater in the US participants compared with participants from the 4 countries (Supplementary Fig. 4a; ANCOM-BC; $q < 0.05$). The non-obese exhibited a significantly greater abundance of genes that catalyze the production of butyrate via the fermentation of pyruvate or branched amino-acids such as enoyl-CoA hydratase enzyme (K01825), Leucine/Acetyl CoA

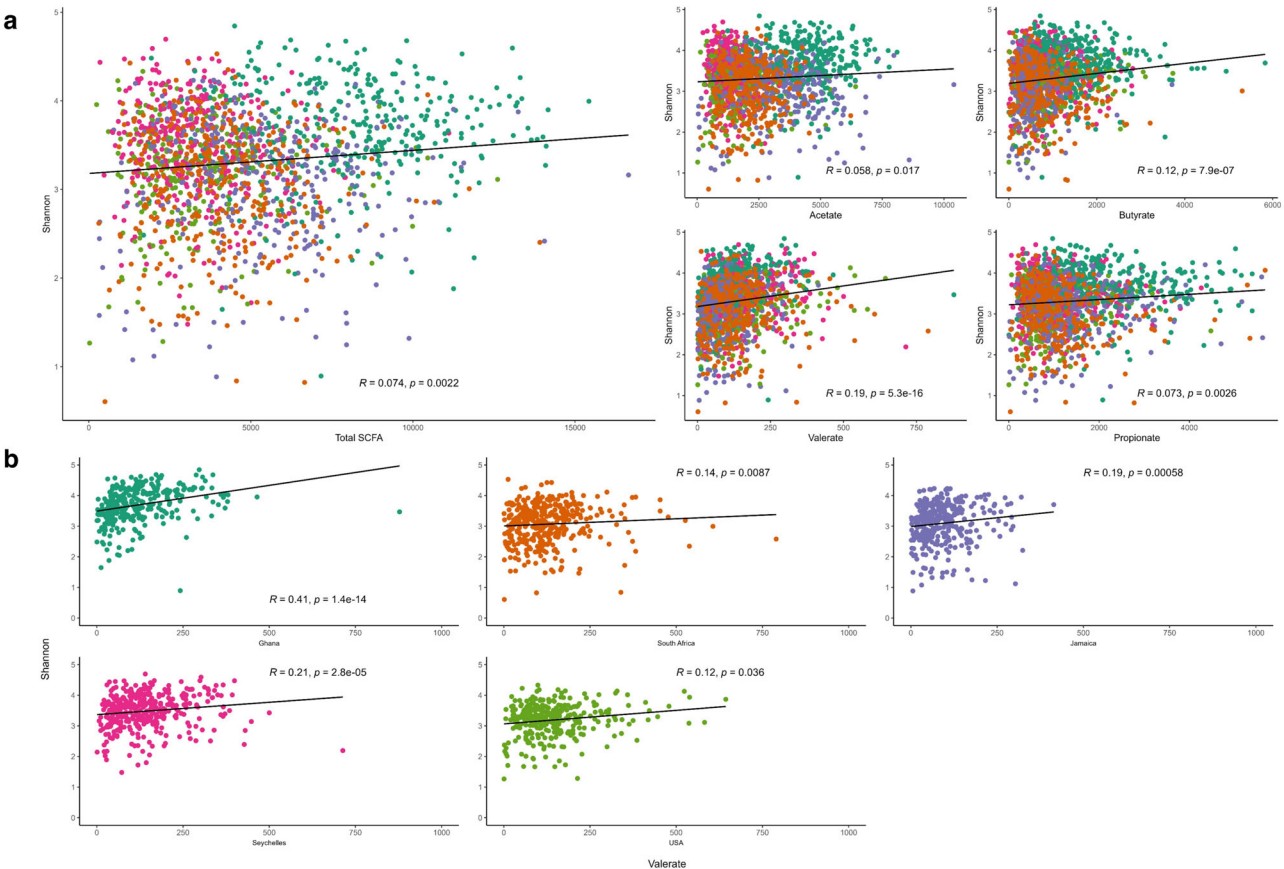

**Fig. 4 | Shannon index correlates positively with fecal short-chain fatty acids.**
**a** Correlations (Spearman's rho, *R*; two-sided) between Shannon diversity and concentrations (*n* = 1704) of the different types of fecal short-chain fatty acids (SCFAs) among countries. **b** Country level correlations (Spearman's rho, *R*; two-sided) between Shannon and valerate levels in Ghana (*n* = 331); South Africa

(*n* = 362); Jamaica (*n* = 331); Seychelles (*n* = 374); US (*n* = 306). Each colored dot represents a sample of a specific country, and the horizontal line on the scatterplot denotes the line of best fit. Unadjusted *p* values are reported. Source data are provided as a Source Data file.

enzyme (K01640) and pyruvate/acetyl CoA enzyme (K00171, K00172, K00169, K01907) by contrast obese individuals were differentially enriched for succinyl-CoA:acetate CoA-transferase (K18118) (Supplementary Fig. 4b; ANCOM-BC; *q* < 0.05). All analyses were adjusted for country, sex, BMI, and age (*q* < 0.05).

Exploring similar analysis at each study site, several predicted genes, including enzymes involved in the pyruvate/acetyl CoA pathway (K01907, K00023, K01640), 4−hydroxybutyrate dehydrogenase (K18120) and 4−hydroxybutyryl−CoA dehydratase (K14534) were differentially enriched in non-obese participants from Ghana (Supplementary Fig. 4c; ANCOM-BC; *q* < 0.05). The relative abundance of two genes both encoding pyruvate/acetyl CoA enzymes (K00171, K00169) was greater among the non-obese South African cohort (Supplementary Fig. 4d; ANCOM-BC; *q* < 0.05), while only a single gene encoding 4−hydroxybutyryl−CoA dehydratase (K14534) was found to be differentially abundant in non-obese individuals from Jamaica (Supplementary Fig. 4e; ANCOM-BC; *q* < 0.05). No statistically significant differences in the proportion of genes encoding enzymes in the butyrate synthesis pathway were observed among participants in the non-obese group compared with the obese counterparts in both Seychelles and the US.

Several gut microbial predicted genes involved in LPS biosynthesis differentially enriched among the countries were identified. In particular, the relative abundance of specific LPS genes (K02560, K12973, K02849, K12979, K12975, K12974) was significantly enriched in Ghana, South Africa, Jamaica, and Seychelles when compared with the US (Supplementary Fig. 5a; ANCOM-BC; *q* < 0.05). Higher proportions of LPS genes, including K12981, K12976 K09953, and K03280, were

significantly increased in Seychelles samples in comparison with US samples and significantly increased in the US cohorts in comparison with participants from Ghana, South Africa, and Jamaica (Supplementary Fig. 5a, ANCOM-BC; *q* < 0.05). US samples had a greater proportion of the following genes (K15669, K09778, K03273, K03271) in comparison with the other 4 countries (Supplementary Fig. 5a; ANCOM-BC; *q* < 0.05). Obese individuals had a greater abundance of predicted genes encoding LPS biosynthesis (K02841, K02843, K03271, K03273, K19353), whereas only 1 LPS gene (K02850) differentially elevated in the non-obese group (Supplementary Fig. 5b; ANCOM-BC; *q* < 0.05). When analyzed separately for each country, the relative proportion of predicted genes encoding components of the LPS biosynthesis was not significantly different between non-obese and obese individuals at all 5 study sites. All analyses were adjusted for country, sex, BMI, and age.

## Microbial community composition and taxonomy correlate with observed fecal SCFA concentrations

Using multiple linear regression analysis, adjusting for age and sex, all countries had significantly higher weight-adjusted fecal total SCFA levels when compared to the US participants (*p* < 0.001), with Ghanaians having the highest weight-adjusted fecal total SCFA levels (Supplementary Table 12). When compared to their obese counterparts, non-obese participants had significantly higher weight-adjusted fecal total and individual SCFA levels (Supplementary Table 13). Total SCFA levels displayed a weak but significantly positive correlation with Shannon diversity (Fig. 4a; Spearman *r* = 0.074). A similar trend was observed in the different individual SCFAs, namely acetate (Fig. 4a;

Spearman $r = 0.058$), butyrate (Fig. 4a; Spearman $r = 0.12$), valerate (Fig. 4a; Spearman $r = 0.19$) and propionate (Fig. 4a; Spearman $r = 0.073$). Observed ASVs were not significantly correlated with total SCFAs ($p > 0.05$). Levels of acetate, butyrate, and propionate exhibited strong significant correlations with total SCFA, whereas valerate levels significantly correlated negatively (Spearman $r = -0.09$) with total SCFAs. Next, we assessed if levels of total SCFAs could be predicted by a mixed model. The country explained 45.7% of the variation in SCFAs. No significant effect was explained either by obesity or Shannon diversity.

When stratified by country, Shannon diversity was negatively associated with acetate concentrations in Jamaica (Supplementary Table 14; Spearman $r = -0.21$, $p < 0.001$) and Seychelles (Supplementary Table 14; Spearman $r = -0.11$; $p = 0.032$), and negatively associated with propionate concentrations in Ghana (Supplementary Table 14; Spearman $r = -0.14$, $p = 0.013$) and South Africa (Supplementary Table 14; Spearman $r = -0.16$, $p = 0.003$). A negative relationship was observed between levels of total SCFA and Shannon in South Africa (Supplementary Table 14; Spearman $r = -0.11$, $p = 0.046$) and Jamaica (Supplementary Table 14; Spearman $r = -0.12$, $p = 0.029$), whereas butyrate correlated with Shannon in South Africa only (Supplementary Table 14; Spearman $r = -0.14$, $p = 0.008$). Finally, valerate levels showed significant correlations with Shannon in all countries (Fig. 4b; Spearman $r = 0.12 – 0.41$, $p < 0.05$).

Using the XGBoost machine learning model, the predictive capacity of SCFAs to stratify individuals to either obese or non-obese in the entire cohort was assessed. The predictive performance of the model was calculated by area under the receiver operating characteristic curve (AUC) analysis, which showed poor accuracy and similar outcomes for the different SCFAs. Total SCFA predicted an obese state with AUC = 0.55, acetate, and propionate with AUC = 0.53, butyrate with AUC = 0.52, and valerate with AUC = 0.51 (Supplementary Fig. 6). Similar analysis to determine obesity status was performed at the country-specific level and the results of the predictive performance of the model were comparable to that of the entire cohort analysis (Supplementary Fig. 6). The comparative predictive capacity for the obese state was higher in Ghana (AUC = 0.60) using valerate; higher in south Africa (AUC = 0.55) using propionate; higher in Jamaica using acetate (AUC = 0.56) and total SCFA (AUC = 0.56). The obese state predicted by butyrate was comparable among all countries (AUC = 0.51) except Ghana (AUC = 0.46). Overall, the predictive capacity of SCFAs was higher in Ghana, South Africa, and Jamaica compared with the US and Seychelles (Supplementary Fig. 6).

Based on the biological plausibility of the associations among the gut microbiota, SCFA, and obesity[7,37,54], and our findings, we applied mediation analysis to evaluate whether the gut microbiota could mediate the relationship between SCFAs and obesity. Our results showed a significant direct effect (ADE) (estimate = −0.0003; $p < 2e-16$) and a total effect of SCFA (estimate = −0.0003; $p < 2e-16$) (Supplementary Table 15). However, the indirect effect (ACME) of SCFA on obesity through Shannon was not statistically significant ($p > 0.05$), suggesting that the effect of SCFA on obesity cannot be fully explained by the microbiota alpha diversity. When the analysis was stratified by country, the effect of SCFA on obesity diminished.

To further explore the connection between SCFAs with gut microbiota, Spearman correlations between taxa that were significantly proportionally different between countries and concentrations of SCFAs were determined. Valerate negatively correlated with the proportion of *Faecalibacterium*, *Roseburia*, and *Streptococcus*, which were all positively correlated with acetate, propionate, and butyrate (Fig. 5a; $q < 0.05$). In addition, the proportions of *Christensenellaceae*, and UCG 002 (*Oscillospiraceae*) were significantly positively associated with valerate and negatively correlated with acetate, propionate, and butyrate (Fig. 5a; $q < 0.05$). Similarly, Spearman's rank correlation coefficients were calculated between the differentially

abundant ASVs identified between obese and non-obese groups with concentrations of SCFAs. Broadly, the proportions of most ASVs were significantly positively associated with acetate in comparison with the other three SCFAs (Fig. 5b; $q < 0.05$). Consistent with the correlations mentioned above, valerate negatively correlated with most ASVs that were found to be positively correlated with the three major SCFAs, acetate, propionate, and butyrate, and vice versa. The relative proportions of ASVs belonging to *Allisonella* and *Lachnospira* positively correlated with acetate, propionate, and butyrate, whereas a significantly negative relationship was observed between *Bacteroides* abundances with the aforementioned SCFAs (Fig. 5b; $q < 0.05$). Valerate showed significantly positive associations with *Oscillospirales* and *Ruminococcaceae* abundances and significantly negative correlations with *Lachnospira* and *Clostridium* abundances (Fig. 5b; $q < 0.05$).

At the country level, several genera contributed to variations in SCFAs. The genera that correlated with SCFA levels for obese and non-obese states differed at each site. For instance, acetate levels correlated negatively with the relative proportions of *Anaerostipes* among obese participants from Ghana and Seychelles and non-obese individuals from Jamaica (Fig. 5c–e; $q < 0.05$). By contrast, positive associations were observed between acetate and *Cantebacterium* among the non-obese US cohort and negative associations with *Coproccocus* within the non-obese Seychelles group (Fig. 5f, g). Butyrate levels positively correlated with 2 different ASVs assigned to the genus *Subdoligranulum* in all countries except the US; ASV 6915 was positively associated with non-obese individuals in Ghana, South Africa, and Jamaica, whereas ASV 7064 was positively associated with obese individuals in Jamaica and Seychelles, which could be indicative of two different species or functional niche differentiation within a taxon. Similarly, 3 *Blautia* ASVs positively correlated with butyrate levels in non-obese participants from Ghana (ASVs 12508, 12561), South Africa (ASVs 12508, 12561), and Jamaica (ASV 12630). Additionally, ASVs 12822 and 12561 positively correlated with butyrate levels in Jamaican participants irrespective of their obesity status (Fig. 5c–f). Propionate was found to be positively associated with *Prevotella* in the non-obese group from Jamaica and Ghana while positively correlated with *Blautia* (ASVs 12822, 12561) and *Coprococcus* (11,293) among obese participants from Seychelles and the US (Fig. 5c–f; $q < 0.05$). In the Ghanaian cohort, valerate positively correlated with *Blautia*, while being inversely associated with *Streptococcus* in the obese group in Jamaica and Seychelles and with all South African participants (Fig. 5c–e; $q < 0.05$). Collectively, more ASVs correlated with total SCFA among non-obese participants when compared to their obese counterparts.

## Discussion

By leveraging a well-characterized large population-based cohort of African-origin adults residing in geographically distinct regions of Ghana, South Africa, Jamaica, Seychelles, and the US, we examined the relationships between gut microbiota, fecal SCFAs, and adiposity. Our data revealed profound variations in gut microbiota, including significant changes in community composition, structure, and predicted functional pathways as a function of population obesity and geography, despite their shared ancestral background. Our data further revealed an inverse relation between fecal SCFA concentrations, microbial diversity, and obesity; importantly, the utility of the microbiota in predicting whether an individual was non-obese or obese differed considerably by country of origin, being marginally better than chance only in Ghana and not predictive for all other countries. Interestingly, only sex was universally predicted at individual sites; while predictive accuracy for diabetes status was only retained for Ghana (AUC = 0.69) and Jamaica (AUC = 0.66); glucose status only in South Africa (AUC = 0.78); and hypertension was only retained for Ghana (AUC = 0.63), suggesting that predicting metabolic disease indicators from the microbiome was impacted by differences in this relationship between countries. Importantly, fecal

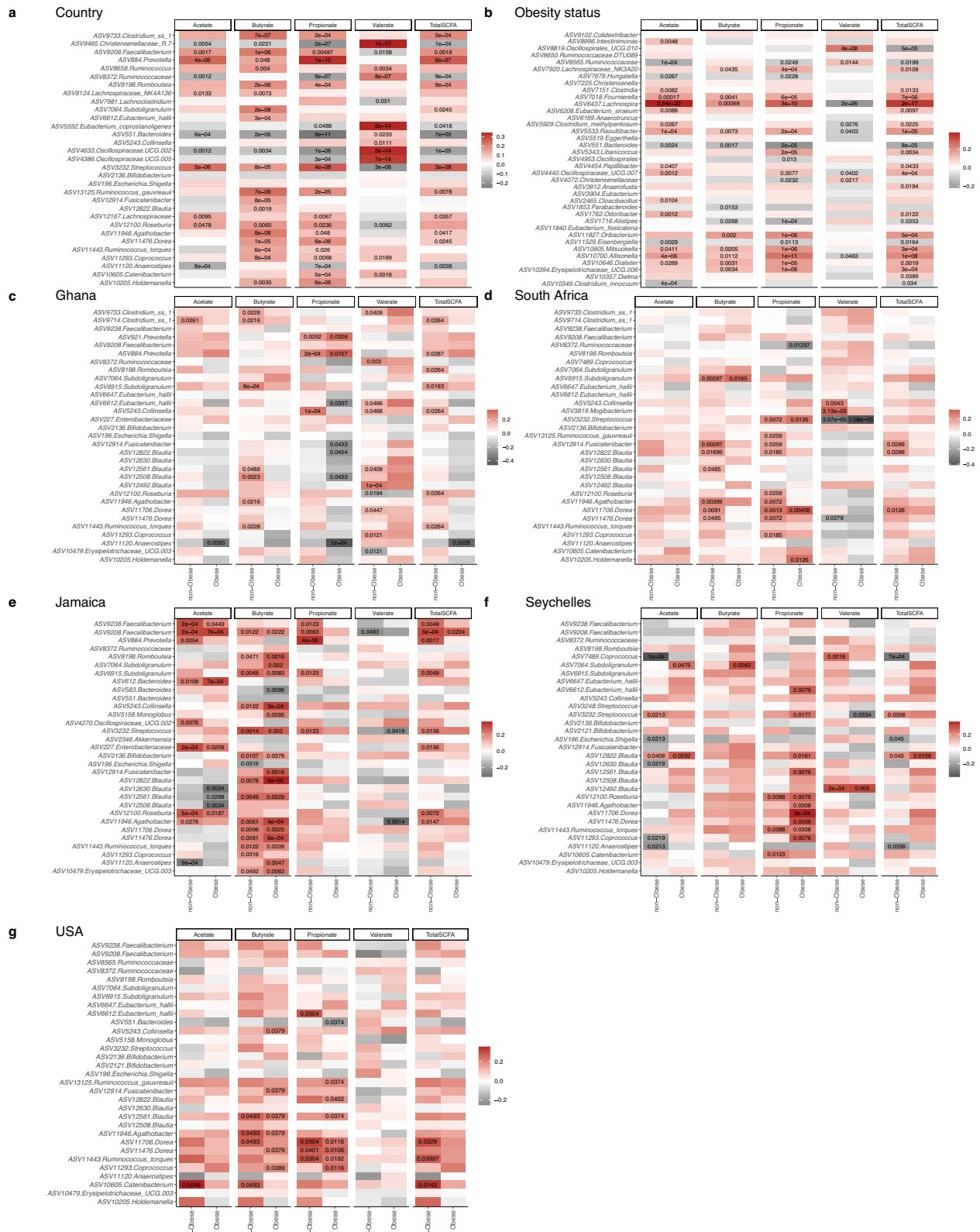

**Fig. 5 | Associations of gut microbiota ASVs with concentrations of short-chain fatty acids (SCFAs). a** Heatmap of Spearman's correlation between concentrations of SCFAs and top 30 differentially abundant ASVs (identified by ANCOM-BC) among countries (*n* = 1694). **b** Heatmap of Spearman's correlation between concentrations of SCFAs and differentially abundant ASVs (identified by ANCOM-BC) for obese (*n* = 1694). Heatmap of Spearman's correlation between concentrations of SCFAs and top 30 relatively abundant features in the non-obese and obese group

in **c** Ghana (*n* = 329); **d** South Africa (*n* = 374); **e** Jamaica (*n* = 386); **f** Seychelles (*n* = 361); **g** USA (*n* = 304). Correlations are identified by Spearman's rank correlation coefficient. Brick red squares indicate positive correlation, gray squares represent negative correlation, and white squares are insignificant correlations. Exact Benjamini–Hochberg adjusted *p* values are shown. Source data are provided as a Source Data file.

SCFA concentrations could not predict obesity either globally or within each, which suggests that the relationship between SCFA and obesity is still unclear, and SCFA may be a poor biomarker for obesity. Overall, our findings are important for understanding the complex relationships between the gut microbiota, population lifestyle, and the development of obesity, which may set the stage for defining the mechanisms through which the microbiome may shape health outcomes in populations of African origin.

As reported previously, our data showed that geographic origin can modulate the composition of the gut microbiota. Our findings are also consistent with our previous METS studies[7,49] and other large-scale continental cohort studies[48,55-61], that report a higher bacterial diversity and composition/microbial richness in traditionally non-western groups that distinguish them from urban-industrialized individuals whose diets are low in fiber and high in saturated fats[62,63]. Although we observe enrichment in the relative abundance of several taxa associated with country of origin in our cohorts, we also detect a pattern where the gut microbiota of the Ghanaian and South African cohorts tend to share many features, while the gut microbiota of the Jamaican cohort shared many features with all 4 countries. It is plausible that Jamaica may be undergoing a more rapid transition than Ghana and South Africa, as reflected by their microbial communities overlapping with both Western and traditionally non-western populations. Notably, traditionally non-western associated taxa, including *Prevotella*, *Butyrivibrio*, *Weisella*, and *Romboutsia*, were enriched in participants from Ghana and South Africa, as suggested previously[61]. Western-associated taxa such as *Bacteroides* and *Parabacteroides* were enriched in individuals from Jamaica and the US[61,64]. We also found greater enrichment of VANISH taxa, including *Butyricicoccus and Succinivibrio*, in the Ghanaian cohort, in line with individuals practicing traditional lifestyles[43]. *Prevotella* is usually associated with plant-based diets rich in dietary fibers, while *Bacteroides* abundance broadly correlates with diets high in fat, animal protein, and sugars[65,66], which is in agreement with our enterotype analysis where a *Prevotella*-rich microbiota dominates the Ghanaian and South African gut, while a Bacteroides-rich microbiota dominated in the high-income countries. As *Prevotella* synthesizes SCFAs[67], its depletion may lead to the observed reduction in SCFA concentrations. Our results support a potential role for geography in reinforcing variations in the gut microbiota in our study cohort despite shared ethnicity. Geography may reflect subtle shifts in lifestyle and/or environmental exposures, including heterogeneity of dietary sources, exposure to medications, socioeconomic factors, medical history, and biogeographical patterns in microbial dispersion[43,44,68,69].

We also inferred the metabolic capacity of the gut microbiota, which suggested that pathways that regulate processes, including energy metabolism, inflammation, epigenetic processes, and oxidative stress, were differently proportional between countries. Participants from Ghana and Jamaica were enriched for gallate degradation, which can result in phenolic catechin metabolites, which are thought to alleviate obesity-related pathologies[70,71]. Additionally, glutamate metabolism, which can be fermented to butyrate and propionate, was enriched in South Africans and Ghanaians compared to the US. In Seychelles, actinobacterial mycothiol biosynthesis was enriched, which is involved in antioxidant activity and the removal of toxic compounds from cells[72]. We further identified a depletion of SCFA synthesis pathways, e.g., acetyl coenzyme A pathway, threonine biosynthesis, and leucine degradation in the US cohort. Further studies are required to evaluate the potential causal relations of these gut microbial functions.

Preclinical mouse models provided early causal links between gut microbial ecology and obesity[73,74], suggesting the potential to predict obesity risk from the microbiome. However, as we showed here, the prediction has proven difficult because the results are conflicting[75]. However, we identified several SCFA-producing bacteria that were significantly depleted in relative abundance among obese individuals, which may influence host energy metabolism. For example, *Oscillospira* and *Christensenella*, which were statistically associated with increased SCFA concentrations and reduced obesity, have previously been associated with a lean phenotype[76-79] and produce SCFAs[77,78], including butyrate, which improves insulin sensitivity and reduces inflammation[80]. We also detected several butyrate-producing ASVs, including *Eubacterium*, *Alistipes*, *Clostridium*, and *Odoribacter*, to be proportionally enriched in individuals who were non-obese. We observed that obese individuals presented a greater abundance of *Lachnospira*, which does produce SCFAs, a finding also consistent with our prior study in the same population[7] and others[81-83]. However, other studies have observed the opposite[84,85].

Some studies, mostly from Western populations, have reported that elevated SCFA concentrations in stool can associate with obesity[37,42,54,86]. For example, a Colombian cohort showed associations between elevated fecal SCFA levels, central obesity, gut permeability, and hypertension[42]. One potential explanation is that obese gut microbiota may lead to less efficient SCFA absorption, hence the increased SCFA excretion[42]. However, we found diets high in fiber correlate positively with weight loss[87,88] and increased levels of fecal SCFAs[89]. One explanation may be differences in lifestyle factors, including medication, activity, and pollutant exposure, which could also impact intestinal absorption in Western countries. We note that fecal SCFA concentrations are not a direct measure of intestinal SCFA production but rather reflect a net result of the difference between production and absorption[90]. Studies using stable isotopes to measure SCFA dynamics would improve the interpretation of this dichotomy.

Our study, due to the size and diversity of the cohort, provides robust evidence to suggest that fecal SCFA concentrations are not predictive of obesity status and that fecal SCFA may function as a poor biomarker for obesity. Previous studies have suggested that measures of both circulating and fecal SCFAs could be more accurate prognostic markers of obesity status[35,42,91,92], a hypothesis that remains to be fully elucidated in our study cohort. Additionally, controlled human intervention studies, including the quantitation of whole-body turnover rates of SCFAs[40], are needed to ascertain the potential health benefits before clinical translation can be implemented to improve metabolic health. Broadly, our findings prompt caution in relying on fecal microbial metabolites alone to infer obesity outcomes since obesity is a heterogenous construct with several unique mechanisms involving host-related factors such as genetic predisposition and microbial SCFAs in precipitating disease susceptibility.

Another mechanism underpinning obesity is metabolic endotoxemia. An increase in Proteobacteria, which often accompanies a high fat/high sugar diet, is often associated with an increase in circulating lipopolysaccharide (LPS) and $H_2S$, which provoke low-grade inflammation, increased intestinal permeability and clock gene disruption in the liver, which associate with adiposity[93-95]. We identified an increase in the LPS producer, *Dialister*[96,97], and LPS-associated pathways in obese individuals, which have previously been associated with obesity[98], sleep disruption, and chronic inflammation[99-103]. We posit that LPS production may result in a systemic inflammatory state favoring the development of obesity in concordance with the associated metabolic endotoxemia pathway linking gut bacteria to obesity.

In obese individuals, as well as SCFA metabolism, we also detected marked depletion in pathways involved in cell structure, vitamin B6, nicotinamide adenine dinucleotide, and amino acid biosynthesis. This suggests that pathways important for growth and energy homeostasis are disrupted in individuals with obesity. We also noted an enrichment of the formaldehyde assimilation I (serine pathway) pathway. A study reported increases in the abundance of formaldehyde assimilation pathway in a depressed group when compared with non-depressed controls[104]. Although we do not yet understand the mechanistic details, it is known that toxic formaldehyde is generated along with

reactive oxygen species during inflammatory processes[105]. Thus, an increased capacity for formaldehyde pathway may indicate a microbiome-induced increase in reactive oxygen species in the gut of obese individuals. Indeed, prior work has identified the induction of oxygen stress by microbial perturbations as one of the mechanisms by which the microbiome can promote weight gain and insulin resistance[106]. The specific alterations of the gut microbiota and the associated predicted functionality may constitute a potential avenue for the development of microbiome-based therapeutics to treat obesity and/or to promote and sustain weight loss.

While our study has several strengths, including a large sample size, a diverse population along an epidemiological transition gradient with a comprehensive dataset that allowed the exclusion of the potential effects of origin as well as control of potential interpersonal covariates, and the use of validated and standard tools for data collection, we acknowledge some limitations as well. First, the cross-sectional nature of our study design is unable to establish temporality or identify mechanisms by which the gut microbiome may causally influence the observed associations. In that regard, we expect that prospective data from the METS cohort study will provide the basis to assess the longitudinal association between gut microbiota composition, metabolites, and obesity, and we have an ongoing study exploring the potential correlations longitudinally. We had no information on diet and physical activity, lifestyle factors that are well-recognized to have profound influences on the gut microbiota. The use of 16 S rRNA sequencing in our analysis for inferences on microbial functional ecology inherently has its limitations for drawing conclusions on species and strain level functionality due to its low resolution. Finally, here we report the associations between the gut microbiota and fecal SCFA concentrations, which may not reflect circulating SCFA concentration levels. Nevertheless, our results provide insight into the relationship between obesity, gut microbiota, and metabolic pathways in individuals of African origin across different geographies, stimulating further examination of large-scale studies using multi-omic approaches with deeper taxonomic and functional resolution and animal transplantation studies to investigate potentially novel microbial strains and to explore the clinical relevance of the observed metabolic differences.

## Methods

### Study cohort
Since 2010, METS and the currently funded METS-Microbiome study have longitudinally followed an international cohort of African origin adults (based on self-report) spanning the epidemiologic transition from Ghana, South Africa, Jamaica, Seychelles, and the US[50,51]. The site in Ghana was based at Nkwantakese, a rural village of approximately 20,000 inhabitants and about 25 km outside of Kumasi. The site in South Africa was in Khayelitsha, an urban informal settlement near Cape Town with over 400,000 inhabitants. The participants from Jamaica were from Spanish Town, an urban area 25 km from the center of Kingston. The Seychelles site was based at Mahé, the largest and most populated of the 100 islands forming the Republic of Seychelles, located approximately 1500 km east of Kenya in the Indian Ocean and home to approximately 81,000 inhabitants. Participants in the US were recruited in Maywood, IL, a suburb adjacent to the western border of Chicago and home to approximately 24,000 people. METS utilizes the framework of the epidemiologic transition to investigate differences in health outcomes based on country of origin. The epidemiologic transition is defined using the United Nations Human Development Index (HDI) as an approximation of the epidemiologic transition. Ghana represents a lower-middle-income country, South Africa represents a middle-income country, Jamaica and Seychelles represent high-income countries, and the US represents a very high-income country. This framework has allowed us to understand how increasing global Westernization, resulting in greater consumption of ultra-processed foods, is associated with a higher prevalence of obesity, type 2 diabetes, and cardiometabolic diseases. Our data from the original METS cohort demonstrate that the epidemiologic transition has altered habitual diets in the international METS sites and that reduced fiber intake is associated with higher metabolic risk, inflammation, and obesity across the epidemiologic transition[107]. Originally, 2,506 African-origin adults (25–45 yrs) were enrolled in METS between January 2010 and December 2011 and followed on a yearly basis. In 2018, METS participants were recontacted and invited to participate in METS-Microbiome (NCT03378765). Participants were excluded from participating in the original METS study if they self-reported being persons with an infectious disease, including HIV, being pregnant, breastfeeding, using antibiotics within 3 months, or having any condition which prevented the individual from participating in normal physical activities. METS-Microbiome was approved by the Institutional Review Board of Loyola University Chicago, IL, US; the Committee on Human Research Publication and Ethics of Kwame Nkrumah University of Science and Technology, Kumasi, Ghana; the Research Ethics Committee of the University of Cape Town, South Africa; the Board for Ethics and Clinical Research of the University of Lausanne, Switzerland; and the Ethics Committee of the University of the West Indies, Kingston, Jamaica. All study procedures were explained to participants in their native languages, and participants were provided written informed consent after being given the opportunity to ask any questions and compensated for their participation.

### Anthropometry, sociodemographic, and biochemical measurements
Participants completed the research visits at the established METS research clinics located in the respective communities[51]. Briefly, they presented themselves at the site-specific research clinic early in the morning, following an overnight fast. The weight of the participant was measured without shoes and dressed in light clothing to the nearest 0.1 kg using a standard digital scale (Seca, SC, USA). Height was measured using a stadiometer without shoes and head held in the Frankfort plane to the nearest 0.1 cm. Waist circumference was measured to the nearest 0.1 cm at the umbilicus, while hip circumference was measured to the nearest 0.1 cm at the point of maximum extension of the buttocks. Adiposity (% body fat) was assessed using bioelectrical impedence analysis (Quantum, RJL Systems, Clinton Township, MI) and study-specific equations[51]. Blood pressure was measured using the standard METS protocol using the Omron Automatic Digital Blood Pressure Monitor (model HEM-747Ic, Omron Healthcare, Bannockburn, IL, USA), with the antecubital fossa at heart level. Metabolic disease risks were assessed as follows: hypertension was defined as mean systolic/diastolic blood pressure ≥130/80 mm Hg or on current treatment; diabetes was defined as >125 mg/dL or current treatment for all sites, except for Ghana as not all participants were fasted overnight. For the Ghanaian site, diabetes was defined as ≥140 mg/dL or current treatment according to American Diabetes Association guidelines for random glucose testing. Obesity was defined as ≥30 kg/m$^2$. Participants were encouraged to provide stool samples in a clinic or just prior to clinic visits using a standard collection kit (EasySampler stool collection kit, Alpco, NH). In cases where this was not possible, participants stored stool samples in home freezers or coolers for 1–3 days prior to clinic visits. Fecal samples were placed within a −80° freezer immediately upon receipt at all the sites. Participants were requested to fast from 8 pm in the evening prior to the clinic examination, during which fasting capillary glucose concentrations were determined using a finger stick (Accu-check Aviva, Roche).

### Fecal short-chain fatty acid quantification
As in our previous studies[7,108–112], fecal SCFAs were measured using LC−MS/MS at the University of Illinois-Chicago Mass Spectrometry Core using previously published methods[113,114]. The LC−MS/MS analysis

was completed on an AB Sciex Qtrap 5500 coupled to the Agilent UPLC/HPLC system. All samples were analyzed by Agilent poroshell 120 EC-C18 Column, 100 Å, 2.7 μm, 2.1 mm × 100 mm coupled to an Agilent UPLC system, which was operated at a flow rate of 400 μl/min. A gradient of buffer A (H$_2$O, 0.1% Formic acid) and buffer B (Acetonitrile, 0.1% Formic acid) was applied as 0 min, 30% of buffer B; increase buffer B to 100% in 4 min; maintain B at 100% for 5 min. The column was then equilibrated for 3 min at 30% B between the injections with the MS detection in negative mode. The MRM transitions of all targeted compounds include the precursor ions and the signature production ion. Unit resolution is used for both analyzers Q1 and Q3. The MS parameters, such as declustering potential, collision energy, and collision cell exit potential, are optimized in order to achieve optimal sensitivity. SCFAs are presented as individual SCFAs (μg/g), including butyric acid, propionic acid, acetic acid, and valeric acid, as well as total SCFAs (sum of 4).

### DNA extraction and amplicon sequencing

Fecal samples were shipped on dry ice to the microbiome core sequencing facility, University of California, San Diego, for 16 S rRNA gene processing. Fecal samples were randomly sorted and transferred to 96-well extraction plates, and DNA was extracted using the MagAttract Power Microbiome kit. Blank controls and ZymoBIOMICS mock controls (Cat. No. D6300) were included per extraction plate, which was carried through all downstream processing steps. Extracted DNA was used for amplification of the V4 region of the 16 S rRNA gene with 515F-806R region-specific primers (515 F: 5'GTGYCAGCMGCCGCGGTAA3'; 806 R: 5'GGACTACNVGGGTWTCTAAT3') according to the Earth Microbiome Project[115,116]. No human DNA sequence depletion or enrichment of microbial or viral DNA was performed. Purified amplicon libraries were sequenced on the Illumina NovaSeq platform to produce 150 bp forward and reverse reads through the IGM Genomics Center at the University of California San Diego.

### Bioinformatic analysis

The generated raw sequence data were uploaded and processed in Qiita[117] (Qiita ID 13512), an open-source, web-enabled microbiome analysis platform. Sequences were demultiplexed, quality filtered, trimmed, erroneous sequences were removed, and ASVs were defined using Deblur[118]. The deblur ASV table was exported to Qiime2[119,120], and representative sequences of the ASVs were inserted into the Greengenes 13.8 99% identity tree with SATé-enabled phylogenetic placement (SEPP) using q2-fragment-insertion[119,121] to generate an insertion tree for diversity computation. Additionally, the deblur ASV table was assigned taxonomic classification using the Qiime2 feature classifier, with Naive Bayes classifiers trained on the SILVA database (version 138;[122]). A total of 463,258,036 reads, 154,952 ASVs, and 1902 samples were obtained from the deblur table. The resulting ASV count table, taxonomy data, insertion tree, and sample metadata were exported and merged into a phyloseq[123] object in R (R Foundation for Statistical Computing, Vienna, Austria) for downstream analysis. Features with less than ten read in the entire dataset, and samples with fewer than 6000 reads were removed from the phyloseq object. In addition, mitochondrial and chloroplast-derived sequences, non-bacterial sequences, as well as ASVs that were unassigned at the phylum level were filtered prior to analyses. There were 433,364,873 reads and 13254 ASVs in the remaining 1873 fecal samples in the phyloseq object. The remaining samples, after filtering, were rarefied to a depth of 6000 reads to avoid sequencing bias before generating diversity measures, leaving 9917 ASVs across 1873 samples.

### Diversity and differential proportional analyses

Alpha diversity measures based on Observed ASVs, Faith's Phylogenetic Diversity, and Shannon Index were conducted on rarified samples using phyloseq v 1.38.0[123] and picante v1.8.2[124] libraries. Beta

diversity was determined using both weighted and unweighted Uni-Frac distance matrices[125], generated in phyloseq v 1.38.0. The *Bacteroides Prevotella* ratio was calculated by dividing the abundance of the genera *Bacteroides* by *Prevotella*. Participants were classified into *Bacteroides* enterotype (B-type) if the ratio was greater than 1. Otherwise, *Prevotella* enterotype (P-type). For differential abundance analysis, samples were processed to remove exceptionally rare taxa. First, the non-rarefied reads were filtered to remove samples with <10,000 reads. Next, ASVs with fewer than 50 reads in total across all samples and/or were present in less than 2% of samples were excluded. This retained 2061 ASVs across 1694 samples. The retained ASVs were binned at the genus level and subsequently used in the analysis of compositions of microbiomes with bias correction (ANCOM-BC v1.2.2)[126] to determine specific taxa differentially abundant across sites or obese phenotype. ANCOM-BC is a statistical approach that accounts for sampling fraction and normalizes the read counts by a process identical to log-ratio transformations while controlling for false discovery rates and increasing power. The site, age, sex, and BMI were added as covariates in the ANCOM-BC formula to reduce the effect of confounders.

### Machine learning

Random Forest supervised learning models implemented in Qiime2 was used to estimate the predictive power of microbial community profiles for the site and obese phenotype. The classifications were done with 500 trees based on 10-fold cross-validation using the QIIME "sample-classifier classify-samples" plugin[120]. A randomly drawn 80% of samples were used for model training, whereas the remaining 20% were used for validation. Further, the 30 most important ASVs for differentiating between site or obese phenotype were predicted and annotated. The predictive capacity of SCFAs to stratify individuals by country or either obese or non-obese was done using XGBoost v.1.7.5.1 machine learning model R package with parameters set at 10-fold CV and 100 repetitions.

### Mediation analysis

The mediation package (v4.5.0) in R (v4.3.0) was used to infer causal relationships between gut microbial diversity, SCFAs, and obesity. The mediation analysis was performed with the same parameter settings (boot = "TRUE," boot.ci.type = "perc", conf.level = 0.95, sims = 1000). The total effect was obtained through the sum of a direct effect and a mediated (indirect) effect.

### Predicted metabolic gene pathway analysis

The functional potential of microbial communities was inferred using the Phylogenetic Investigation of Communities by Reconstruction of Unobserved States 2 (PICRUSt2) v2.5.1 with the ASV table processed to remove exceptionally rare taxa and the representative sequences as input files[127]. The metabolic pathway from the PICRUSt2 pipeline was annotated using the MetaCyc database[128]. The predicted MetaCyc abundances (unstratified pathway abundances) were analyzed with ANCOM-BC to determine differentially abundant pathway associations across sites and obese status. The site, age, sex, and BMI were added as covariates in the ANCOM-BC formula to reduce the effect of confounders.

### Statistical analysis

All statistical analyses and graphs were done with R v4.1.1 and Stata v5. Descriptive statistics for continuous variables are presented as mean ± standard deviation of the mean if normally distributed or as median (interquartile range) if non-normally distributed; categorical variables are shown as counts and percentages. P values were two-sided. Kruskal–Wallis test and Permutational Analysis of Variance (PERMANOVA) tests with 999 permutations using the Adonis function in the vegan package v 2.6.2[129] were performed to compare alpha and

beta diversity measures, respectively, with multiple groups comparison correction. PERMANOVA models were adjusted for BMI, age, and sex for the country, whereas age, sex, and country were accounted for in obese groups. For variables that showed significant differences in the PERMANOVA analyses, the PERMDISP test was performed to assess differences in dispersion or centroids. For differential abundance analysis, the false-discovery rate (FDR) method incorporated in the ANCOM-BC library was used to correct $p$-values for multiple testing. A cut-off of $q < 0.05$ was used to assess significance. Spearman correlations were performed between concentrations of short-chain fatty acids, Shannon diversity or concentrations of short-chain fatty acids, and differentially abundant taxa that were identified either among study sites or in obese and non-obese individuals. The resulting p-values were adjusted for multiple testing using the Benjamini–Hochberg FDR. A mixed model was built using lme4 package vk1-31 to assess whether total SCFAs could be predicted by Shannon diversity, obesity, and country, setting obesity and Shannon diversity as fixed effects and random intercept by country. Statistical analysis and data visualization performed in R version 4.1.1 used the following freely available packages: ANCOMBC v1.2.2, ggplot2 v 3.3.6, vegan v 2.6.2, phyloseq v 1.38.0, microbiome v 1.19.1, microbiomeutilities v 1.00.16, gghalves v 0.1.3, qiime2R v0.99.6, tidyverse v1.3.1, reshape v0.8.8, microViz v 0.9.4, cowplot v1.1.1, picante v1.8.2, reshape v 0.8.8, lme4 v1.1-31, RColorBrewer v1.1-3, gtable v 0.3, mediation v4.5.0, xGBoost v1.7.5.1, Biostrings v2.62.0, biomformat v1.22.0, rstatix 0.7.0, patchwork 1.1.1, readr v2.1.2. PICRUSt2 v2.5.1 was installed in Python v 3.7.4.

### Reporting summary
Further information on research design is available in the Nature Portfolio Reporting Summary linked to this article.

## Data availability
All 16 S rRNA gene sequence data have been deposited at the European Bioinformatics Institute site under the accession code (https://www.ebi.ac.uk/ena/browser/view/PRJEB63378). Additionally, sequencing data and processed tables are available through QIITA[117] under study identifier 13512. The SILVA 16 S rRNA database used for alignment is available at https://data.qiime2.org/2022.2/common/silva-138-99-515-806-nb-classifier.qza. The KEGG and MetaCyc Databases are available at https://www.genome.jp/kegg/ and https://metacyc.org/, respectively. The clinical and metadata are available under restricted access due to privacy regulations of our cohort, access can be obtained by request to the corresponding author (Lara Dugas: ldugas@luc.edu). The data and analyses generated in this study are available within the paper, Supplementary Information and Source Data files provided with this paper. Source data are provided in this paper.

## Code availability
The R codes used for analysis and figure generation are available at https://doi.org/10.6084/m9.figshare.23542395.v1.

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

## Acknowledgements

We thank the METS participants who continue their ongoing participation in the METS studies, as well as the site-specific clinic staff in Ghana, South Africa, Jamaica, Seychelles, and the US. The UCSD Microbiome Core performed sample extractions and library preparation utilizing protocols and primers published on the Earth Microbiome Project website (https://earthmicrobiome.org/). This publication includes data generated at the UC San Diego IGM Genomics Center utilizing an Illumina NovaSeq 6000 that was purchased with funding from a National Institutes of Health SIG grant under Award Number S10 OD026929 (J.A.G.). This work was supported by the National Institutes of Health grant under Award Number R01-DK111848 (L.R.D.).

## Author contributions

L.R.D. and B.T.L. conceived the study. L.R.D., C.C.-K., P.B., B.V., K.B.-A., J.P.-R., P.O.B., T.E.F., M.W., E.V.L., D.R., N.S., S.O., and A.L. collected human samples and metadata. G.E.-M. and C.C.-K. curated metadata. S.D. performed sequencing of samples. G.E.-M., C.C.-K., and M.G.M. conducted formal analysis and visualization. J.A.G. and L.R.D. supervised and provided feedback on formal analysis and visualization. G.E.-M., C.C.-K., L.R.D., and J.A.G. wrote the original paper. L.R.D. and J.A.G. secured the funding. All authors edited and approved the final paper.

## Competing interests

The authors declare no competing interests.
