## [Peer Review File · Nature Communications]

REVIEWER COMMENTS

Reviewer #1 (Remarks to the Author):

Summary

Eklun-Mensah, Choo-Kang et al. present results from a cross-sectional analysis of 16S amplicon sequencing data, fecal SCFAs, and obesity-related measures covering 5 African origin cohorts from distinct geographical regions. The authors inferred associations of microbial diversity indices, amplicon sequence variants, and predicted metabolic capacity with obesity and SCFA abundances. In particular, they show that the relationship between gut microbial composition and obesity is heterogeneous across the cohorts. I think the study is well-designed and covers cohorts that are severely underrepresented in the existing literature (<https://doi.org/10.1371/journal.pbio.3001536>).

I feel that the analysis could be improved, however, because it often pools information across the cohorts, heavily confounds clinical measures with geographical differences, and is lacking some more nuance on phylogeny of the microbial taxa covered in the study. Making better use of the balanced cohort design and opting for a stratified analysis strategy in favor of pooled analyses is likely to improve the manuscript significantly.

Suggested major changes

As mentioned above, I think stratifying analyses like the ones presented in Fig. 2-5 by cohort/country would make better use of the study design. I would suggest studying the microbiome associations with SCFA abundance and anthropogenic measures within each cohort first, and then reporting the observed heterogeneity in the observed associations. This is also what is suggested by the current NIH guidelines when studying cohorts that span several ethnic groups ("Reducing Bias and Examining Intervention Effects" at <https://grants.nih.gov/policy/inclusion/women-and-minorities/analyses.htm>). Luckily, the study is in a superb spot to do so due to the balanced cohorts and sufficiently large numbers of individuals in each of the 5 cohorts.

In Figure 2, why was the US population chosen as a reference group? I feel that this figure would be more informative by showing the information in panel b) but stratified by country followed by a discussion of overlap and differences between the obesity-associated ASVs.

As the authors observe, gut microbial composition heavily differs across cohorts from different geographic regions, which is particularly apparent in 16S studies that are limited to a small highly variable region of bacterial genomes. Thus, most of the identified ASVs will only be observed in one particular cohort, and only few ASVs will be shared across all cohorts. This creates a block structure in ASVs abundance matrixes that will be readily exploited by any machine learning or regression methods and lead to inflated accuracy metrics as observed by the authors in Fig. 3a. This will also bleed in to all other measures that are correlated with the country of origin, such as the outcomes studied in Fig. 3b-f and those confounding effects are hard to get rid of in random forest models. I would expect that the models in Fig. 3b-f would not perform well within each cohort.

In that vein, I found the stratified random forests model in Supplemental Figure 3 the most interesting result in the paper. This was prominently mentioned in the abstract, so it is a bit surprising that it is buried in the supplement. The authors state that microbiome-based prediction performance of anthropogenic measures is related to the epidemiological transition, but I wonder if this might also be confounded by alpha diversity where the RF models have access to more features in some of the cohorts due to the larger number of unique ASVs.

Apart from stratifying the analyses in Fig. 5 by country I would also have liked to see how those ASV-SCFA associations relate to the phylogenetic distance of ASVs in some larger taxonomic group, like the bacterial genus. For instance, are bacterial genera associated with SCFA levels across all cohorts more phylogenetically conserved? In case a genus is associated with SCFA production in only one cohort, have the ASVs in this cohort diverged from their respective genus-associated ASVs in the other cohorts? This would allow pinpointing whether there is a signal of potential adaptation in the specific cohorts.

Since the goal of the study seems to be assessing the relationship between the gut microbiome, SCFA production, and obesity, it is probably missing a figure showing the relationship between SCFA production and obesity stratified by country. As far as I am aware, most of the evidence for lower SCFA abundances/production rates in obese individuals comes from white affluent populations, so it would be interesting whether this relationship is present or absent in the communities studied here. In case the authors are exclusively interested in the relationship between the two through the gut microbiome, mediation analysis would probably have been the better approach.

The description of the cohort recruitment could be improved. For instance, it is unclear whether individuals within a country were recruited from a distinct geographic region or were sampled representatively across the general population. It is also unclear how the status of "African origin" was determined. What was the exact definition used here and was that assessment made by the recruiting researcher or the study participants themselves?

Suggested minor changes

Figure 1g does not show the correct data. There are more than 13,000 ASVs, but the figure lists less than 100. So either the figure or the caption is incorrect.

Was diet assessed in the study? Because lines 550-553 make it look like it was. If yes, this data should be included in the manuscript and be used to assess whether variation in the microbiome-SCFA/adiposity associations were modulated by dietary patterns.

Fig. 4 should also report the correlations for each separate cohort.

During the placement in the GreenGenes tree and the PiCrust analysis it should be noted that ASVs from understudied populations are approximated by full genome sequences from overrepresented populations here, which has some inherent bias because the used proxy genomes are unlikely to be a faithful representation of what is actually present in the studied communities (for instance see <https://doi.org/10.1016/j.cell.2021.02.052>). I don't think either adds that much to the manuscript. So I would suggest to either rely on direct measures derived from the ASV sequences at hand or to add a disclaimer to the manuscript that mentions this bias.

As far as I am aware, DEBLUR only uses the forward reads. Was the reported number of reads corrected for that? Why was a paired-end sequencing protocol used even if the reverse reads were omitted in later analyses?

Line 160-162: This should read "a total of 433,364,873 16S V4 amplicon sequences were generated".

Line 298: Typo in the reported correlation "(r = 0.0.074)".

The discussion is a bit excessive relative to the fairly succinct results section, so I would suggest pruning this to a few selected major points.

Where p-values are reported, that should be accompanied by naming the used test and the effect size.

Strong points of the manuscript

Reviews can be discouraging, especially to early career scientists. Thus, to end the review on a positive note, here is a list of parts where I think the study and manuscript were exceptionally well-crafted.

The introduction makes a great job at summarizing a large body of research and pointing out the challenges.

The study cohort was excellently assembled, included local institutions and stakeholders, and has added them as authors. Very few studies have that level of awareness for social equity.

I love how the authors made sure that all samples were processed in few batches, at the same site and with consistent methodology.

Even though fecal SCFAs have been measured before, few studies make the efforts to do so for samples from diverse populations, as this requires transporting volatile samples over large distances or introducing biases due to the different processing sites. The standardized measures here will be of interest to many researchers.

I appreciate that all the data from the manuscript was provided and was accessible during review.

Reviewer #2 (Remarks to the Author):

In this manuscript, the authors leverage the METS cohort – a well-phenotyped multi-country study of adults of African origin – to explore associations between the gut microbiome, short-chain fatty acids (SCFA), and obesity. The authors profiled the microbiome (using 16S sequencing) and levels of SCFA in >1,800 fecal samples – a remarkable effort. The profiling was done according to the best practices in the field. Based on these alone, this is a remarkable resource, and it would be great to see it shared with the scientific community. From a biological perspective, progress in understanding the predictive utility of the microbiome in various pathologies and its role in their manifestation has been hampered by the under-representation of diverse populations. The METS cohort, which profiles individuals from countries representing a broad spectrum of income levels, offers an attractive opportunity to test whether the microbiome (and its metabolites) are universal markers of obesity. The authors do reach a conclusion regarding the predictive ability of the microbiome. However, I would suggest a closer look at the interpretation (see below). I would also like to see the discussion of the role of SCFA expanded further (see below). Beyond these two points, I have some comments regarding the study design that I would like to see the authors address in the text. Overall, a remarkable achievement!

Main comments

A. Microbiome studies focused on non-US, non-Western Europeans are critically needed to understand whether a given disease is associated with a universal microbiome signature (that can be thus harnessed for prediction or can be targeted) or, rather, distinct populations exhibit unique markers for the same disease. From this perspective, this study design is suitable to address this important question in the context of obesity. The authors' conclusion is that "...microbiota differences between obesity and non-obesity may be larger in low-to-middle-income countries compared to high-income countries", and again in the discussion, "the utility of the microbiota in predicting whether an individual was lean or obese was inversely correlated with the income-level of the country of origin.". I have summarized in the table below the authors' findings. I am finding it difficult to find support for this claim in the data. Observed ASVs and unweighted beta diversity partly support this claim, but the weighted beta diversity and the area under the receiver-operator curve do not. Notably, obesity is best predicted by the microbiome by Ghana (low income), followed by the US (highest income) – this does not follow the epidemiological transition. This is such an important take-home message that I think deserves more attention. I agree with the authors regarding a potential underlying cause for the discrepancy between weighted and unweighted beta diversity, but this does not explain why the AUROC does not follow the epidemiological transition. The authors should try to identify the factors that allow better prediction in Ghana, South Africa, and the US, but not Jamaica and Seychelles. Even if no such factors are identified, the conclusions of the analyses should be adjusted to better represent the data, throughout the manuscript.

Ghana South Africa Jamaica Seychelles US

Observed ASVs Sig. Sig. NS NS NS

Weighted BDiv NS NS NS NS Sig.

Unweighted BDiv Sig. Sig. NS NS NS

AUROC 0.75 0.61 0.49 0.57 0.64

AUROC Rank 1 3 5 4 2

AUROC Trend

B. The manuscript aims to address an important open question in the field of microbiome and metabolic syndrome – whether higher levels of SCFAs are associated with improved or poorer metabolic health (in this case, obesity). The authors find that “When compared to their obese counterparts, non-obese participants had significantly higher weight-adjusted fecal total and individual SCFA levels”. What I’m still missing here is: A) can obesity be predicted according to SCFA levels (total, and each SCFA) – similar to what the authors achieved with the microbiome? B) Are there differences between the countries in the ability of SCFA to predict obesity? This would be such a valuable finding, regardless of the result.

C. Study design. Unclear if antibiotic treatment was an exclusion criterion, I could not find any indication of that. Please address that directly in the methods. If antibiotics use was not an exclusion criterion, please address in your discussion how this might have impacted the results, or preferably, validate the results after omitting antibiotics-associated samples, if data for this purpose are available.

D. Study design. Unclear how stool samples were kept before transport to the research clinic (home refrigerator, freezer?), how were they transferred (coolers? Room temperature?), what was the maximum allowed age of sample (assuming not all patients had a sample the day of the visit – how many days of storage were allowed?). The age of the sample, and the temperature in which it was stored and transported can significantly impact the microbiome, metabolites, volatiles, and SCFA. Please add relevant information to the methods. Please address in the discussion how this might have impacted the results.

Additional comments

1. Abstract, “lowest and highest end of the epidemiologic transition spectrum, respectively” – It might be worthwhile to clarify what the “epidemiological transition spectrum” means or use alternative, more common terms. It’s defined in the introduction but complicates interpretation of the abstract.
2. Introduction, “A major driver of obesity is the adoption of a western lifestyle, which is characterized by excessive consumption of ultra-processed foods.” – please add a reference here.
3. Introduction, regarding the conflicting evidence on SCFA in obesity, it might be worthwhile to mention Perry, Nature 2016; Tirosh, Sci Transl Med 2019.
4. Introduction, Line 50-51 – I advise adjusting the language here when discussing deaths related to obesity. It is not simply ‘obesity’ that accounts for 60% of deaths related to high body mass index, as obesity is describing a high body mass index. Rather, it is comorbidities associated with obesity that are accounting for these deaths.
5. Methods, “HIV-positive individuals”, please use person-first language.
6. Methods, Please include registration numbers for the trials.
7. Methods, please consider including the STORMS microbiome checklist:
<https://www.stormsmicrobiome.org/>
8. Methods, “METS data showed Ghanaians consumed the greatest amount...”, why is this section here?
9. Methods, lines 498-500, this sentence is unclear.
10. When citing panels from figures in text, please denote which panel in the figure is being discussed.
11. Bacterial family names should be italicized.
12. Figure 3 - for ease of understanding, please title each graph with the variable that is being modeled. Additionally, is there any significance to the difference between macro- and micro-average in some of the ROC curves? For example, diabetes status (Fig. 3C) seems to be well explained by the macro-average as compared to the micro-average. Also, for these models, a confusion matrix might be a good supplemental figure to provide for classifier metrics and additional understanding to the readers.
13. ANCOM-BC is referred to as ‘ANCOM-BC’ and ‘ANCOMBC’ in the text; this should be ‘ANCOM-BC’ and consistent throughout.

14. Supplemental Figures 5 and 6 - it would be worth having the pathway name in addition to or replacement of the KO pathway number.

NCOMMS-23-15272. Gut microbiota and fecal short chain fatty acids differ with adiposity and country of origin: The METS-Microbiome Study

REVIEWERS COMMENTS

Reviewer #1 (Remarks to the Author):

Summary

Ecklu-Mensah, Choo-Kang et al. present results from a cross-sectional analysis of 16S amplicon sequencing data, fecal SCFAs, and obesity-related measures covering 5 African origin cohorts from distinct geographical regions. The authors inferred associations of microbial diversity indices, amplicon sequence variants, and predicted metabolic capacity with obesity and SCFA abundances. In particular, they show that the relationship between gut microbial composition and obesity is heterogeneous across the cohorts. I think the study is well-designed and covers cohorts that are severely underrepresented in the existing literature (<https://doi.org/10.1371/journal.pbio.3001536>).

I feel that the analysis could be improved, however, because it often pools information across the cohorts, heavily confounds clinical measures with geographical differences, and is lacking some more nuance on phylogeny of the microbial taxa covered in the study. Making better use of the balanced cohort design and opting for a stratified analysis strategy in favor of pooled analyses is likely to improve the manuscript significantly.

Suggested major changes

As mentioned above, I think stratifying analyses like the ones presented in Fig. 2-5 by cohort/country would make better use of the study design. I would suggest studying the microbiome associations with SCFA abundance and anthropogenic measures within each cohort first, and then reporting the observed heterogeneity in the observed associations. This is also what is suggested by the current NIH guidelines when studying cohorts that span several ethnic groups (“Reducing Bias and Examining Intervention Effects” at <https://grants.nih.gov/policy/inclusion/women-and-minorities/analyses.htm>). Luckily, the study is in a superb spot to do so due to the balanced cohorts and sufficiently large numbers of individuals in each of the 5 cohorts

In Figure 2, why was the US population chosen as a reference group? I feel that this figure would be more informative by showing the information in panel b) but stratified by country followed by a discussion of overlap and differences between the obesity-associated ASVs.

As the authors observe, gut microbial composition heavily differs across cohorts from different geographic regions, which is particular apparent in 16S studies that are limited to a small highly variable region of bacterial genomes. Thus, most of the identified ASVs will only be observed in one particular cohort, and only few ASVs will be shared across all cohorts. This creates a block structure in ASVs abundance matrixes that will be readily exploited by any machine learning or

regression methods and lead to inflated accuracy metrics as observed by the authors in Fig. 3a. This will also bleed into all other measures that are correlated with the country of origin, such as the outcomes studied in Fig. 3b-f and those confounding effects are hard to get rid of in random forest models. I would expect that the models in Fig. 3b-f would not perform well within each cohort.

In that vein, I found the stratified random forests model in Supplemental Figure 3 the most interesting result in the paper. This was prominently mentioned in the abstract, so it is a bit surprising that it is buried in the supplement. The authors state that microbiome-based prediction performance of anthropogenic measures is related to the epidemiological transition, but I wonder if this might also be confounded by alpha diversity where the RF models have access to more features in some of the cohorts due to the larger number of unique ASVs.

Apart from stratifying the analyses in Fig. 5 by country I would also have liked to see how those ASV-SCFA associations relate to the phylogenetic distance of ASVs in some larger taxonomic group, like the bacterial genus. For instance, are bacterial genera associated with SCFA levels across all cohorts more phylogenetically conserved? In case a genus is associated with SCFA production in only one cohort, have the ASVs in this cohort diverged from their respective genus-associated ASVs in the other cohorts? This would allow pinpointing whether there is a signal of potential adaptation in the specific cohorts.

Since the goal of the study seems to be assessing the relationship between the gut microbiome, SCFA production, and obesity, it is probably missing a figure showing the relationship between SCFA production and obesity stratified by country. As far as I am aware, most of the evidence for lower SCFA abundances/production rates in obese individuals comes from white affluent populations, so it would be interesting whether this relationship is present or absent in the communities studied here. In case the authors are exclusively interested in the relationship between the two through the gut microbiome, mediation analysis would probably have been the better approach.

The description of the cohort recruitment could be improved. For instance, it is unclear whether individuals within a country were recruited from a distinct geographic region or were sampled representatively across the general population. It is also unclear how the status of "African origin" was determined. What was the exact definition used here and was that assessment made by the recruiting researcher or the study participants themselves?

Suggested minor changes.

Figure 1g does not show the correct data. There are more than 13,000 ASVs, but the figure lists less than 100. So either the figure or the caption is incorrect.

Was diet assessed in the study? Because lines 550-553 make it look like it was. If yes, this data should be included in the manuscript and be used to assess whether variation in the microbiome-SCFA/adiposity associations were modulated by dietary patterns.

Fig. 4 should also report the correlations for each separate cohort.

During the placement in the GreenGenes tree and the PiCrust analysis it should be noted that ASVs from understudied populations are approximated by full genome sequences from overrepresented populations here, which has some inherent bias because the used proxy genomes are unlikely to be a faithful representation of what is actually present in the studied communities (for instance see <https://doi.org/10.1016/j.cell.2021.02.052>). I don't think either adds that much to the manuscript. So I would suggest to either rely on direct measures derived from the ASV sequences at hand or to add a disclaimer to the manuscript that mentions this bias.

As far as I am aware, DEBLUR only uses the forward reads. Was the reported number of reads corrected for that? Why was a paired-end sequencing protocol used even if the reverse reads were omitted in later analyses?

Line 160-162: This should read "a total of 433,364,873 16S V4 amplicon sequences were generated".

Line 298: Typo in the reported correlation "(r = 0.0.074)".

The discussion is a bit excessive relative to the fairly succinct results section, so I would suggest pruning this to a few selected major points.

Where p-values are reported, that should be accompanied by naming the used test and the effect size.

Strong points of the manuscript

Reviews can be discouraging, especially to early career scientists. Thus, to end the review on a positive note, here is a list of parts where I think the study and manuscript were exceptionally well-crafted. The introduction makes a great job at summarizing a large body of research and pointing out the challenges. The study cohort was excellently assembled, included local institutions and stakeholders, and has added them as authors. Very few studies have that level of awareness for social equity. I love how the authors made sure that all samples were

processed in few batches, at the same site and with consistent methodology. Even though fecal SCFAs have been measured before, few studies make the efforts to do so for samples from diverse populations, as this requires transporting volatile samples over large distances or introducing biases due to the different processing sites. The standardized measures here will be of interest to many researchers. I appreciate that all the data from the manuscript was provided and was accessible during review.

Reviewer #2 (Remarks to the Author):

In this manuscript, the authors leverage the METS cohort – a well-phenotyped multi-country study of adults of African origin – to explore associations between the gut microbiome, short-chain fatty acids (SCFA), and obesity. The authors profiled the microbiome (using 16S sequencing) and levels of SCFA in >1,800 fecal samples – a remarkable effort. The profiling was done according to the best practices in the field. Based on these alone, this is a remarkable resource, and it would be great to see it shared with the scientific community. From a biological perspective, progress in understanding the predictive utility of the microbiome in various pathologies and its role in their manifestation has been hampered by the under-representation of diverse populations. The METS cohort, which profiles individuals from countries representing a broad spectrum of income levels, offers an attractive opportunity to test whether the microbiome (and its metabolites) are universal markers of obesity. The authors do reach a conclusion regarding the predictive ability of the microbiome. However, I would suggest a closer look at the interpretation (see below). I would also like to see the discussion of the role of SCFA expanded further (see below). Beyond these two points, I have some comments regarding the study design that I would like to see the authors address in the text. Overall, a remarkable achievement!

Main comments

A. Microbiome studies focused on non-US, non-Western Europeans are critically needed to understand whether a given disease is associated with a universal microbiome signature (that can be thus harnessed for prediction or can be targeted) or, rather, distinct populations exhibit unique markers for the same disease. From this perspective, this study design is suitable to address this important question in the context of obesity. The authors' conclusion is that "...microbiota differences between obesity and non-obesity may be larger in low-to-middle-income countries compared to high-income countries", and again in the discussion, "the utility of the microbiota in predicting whether an individual was lean or obese was inversely correlated with the income-level of the country of origin.". I have summarized in the table below the authors' findings. I am finding it difficult to find support for this claim in the data. Observed ASVs and unweighted beta diversity partly support this claim, but the weighted beta diversity and the area under the receiver-operator curve do not.

Notably, obesity is best predicted by the microbiome by Ghana (low income), followed by the US (highest income) – this does not follow the epidemiological transition. This is such an important take-home message that I think deserves more attention. I agree with the authors regarding a potential underlying cause for the discrepancy between weighted and unweighted beta diversity, but this does not explain why the AUROC does not follow the epidemiological transition.

The authors should try to identify the factors that allow better prediction in Ghana, South Africa, and the US, but not Jamaica and Seychelles. Even if no such factors are identified, the conclusions of the analyses should be adjusted to better represent the data, throughout the manuscript.

Ghana South Africa Jamaica Seychelles US

Observed ASVs Sig. Sig. NS NS NS

Weighted BDiv NS NS NS NS Sig.

Unweighted BDiv Sig. Sig. NS NS NS

AUROC 0.75 0.61 0.49 0.57 0.64

AUROC Rank 1 3 5 4 2

AUROC Trend

B. The manuscript aims to address an important open question in the field of microbiome and metabolic syndrome – whether higher levels of SCFAs are associated with improved or poorer metabolic health (in this case, obesity). The authors find that “When compared to their obese counterparts, non-obese participants had significantly higher weight-adjusted fecal total and individual SCFA levels”. What I’m still missing here is: A) can obesity be predicted according to SCFA levels (total, and each SCFA) – similar to what the authors achieved with the microbiome? B) Are there differences between the countries in the ability of SCFA to predict obesity? This would be such a valuable finding, regardless of the result.

C. Study design. Unclear if antibiotic treatment was an exclusion criterion, I could not find any indication of that. Please address that directly in the methods. If antibiotics use was not an exclusion criterion, please address in your discussion how this might have impacted the results, or preferably, validate the results after omitting antibiotics-associated samples, if data for this purpose are available.

D. Study design. Unclear how stool samples were kept before transport to the research clinic (home refrigerator, freezer?), how were they transferred (coolers? Room temperature?), what was the maximum allowed age of sample (assuming not all patients had a sample the day of the visit – how many days of storage were allowed?). The age of the sample, and the temperature in which it was stored and transported can significantly impact the microbiome, metabolites, volatiles, and SCFA. Please add relevant information to the methods. Please address in the discussion how this might have impacted the results.

Additional comments

- 1. Abstract, “lowest and highest end of the epidemiologic transition spectrum, respectively” – It might be worthwhile to clarify what the “epidemiological transition spectrum” means or use alternative, more common terms. It’s defined in the introduction but complicates interpretation of the abstract.*
- 2. Introduction, “A major driver of obesity is the adoption of a western lifestyle, which is characterized by excessive consumption of ultra-processed foods.” – please add a reference here.*
- 3. Introduction, regarding the conflicting evidence on SCFA in obesity, it might be worthwhile to mention Perry, Nature 2016; Tirosh, Sci Transl Med 2019.*
- 4. Introduction, Line 50-51 – I advise adjusting the language here when discussing deaths related to obesity. It is not simply ‘obesity’ that accounts for 60% of deaths related to high body mass index, as obesity is describing a high body mass index. Rather, it is comorbidities associated with obesity that are accounting for these deaths.*
- 5. Methods, “HIV-positive individuals”, please use person-first language.*
- 6. Methods, Please include registration numbers for the trials.*
- 7. Methods, please consider including the STORMS microbiome checklist: <https://www.stormsmicrobiome.org/>*
- 8. Methods, “METS data showed Ghanaians consumed the greatest amount...”, why is this section here?*
- 9. Methods, lines 498-500, this sentence is unclear.*
- 10. When citing panels from figures in text, please denote which panel in the figure is being discussed.*
- 11. Bacterial family names should be italicized.*
- 12. Figure 3 - for ease of understanding, please title each graph with the variable that is being modeled. Additionally, is there any significance to the difference between macro- and micro-average in some of the ROC curves? For example, diabetes status (Fig. 3C)*

seems to be well explained by the macro-average as compared to the micro-average. Also, for these models, a confusion matrix might be a good supplemental figure to provide for classifier metrics and additional understanding to the readers.

13. ANCOM-BC is referred to as 'ANCOM-BC' and 'ANCOMBC' in the text; this should be 'ANCOM-BC' and consistent throughout.

14. Supplemental Figures 5 and 6 - it would be worth having the pathway name in addition to or replacement of the KO pathway number.

Point by point response to reviewers comments.

Reviewer #1 (Remarks to the Author):

Summary

Ecklu-Mensah, Choo-Kang et al. present results from a cross-sectional analysis of 16S amplicon sequencing data, fecal SCFAs, and obesity-related measures covering 5 African origin cohorts from distinct geographical regions. The authors inferred associations of microbial diversity indices, amplicon sequence variants, and predicted metabolic capacity with obesity and SCFA abundances. In particular, they show that the relationship between gut microbial composition and obesity is heterogeneous across the cohorts. I think the study is well-designed and covers cohorts that are severely underrepresented in the existing literature (<https://doi.org/10.1371/journal.pbio.3001536>).

I feel that the analysis could be improved, however, because it often pools information across the cohorts, heavily confounds clinical measures with geographical differences, and is lacking some more nuance on phylogeny of the microbial taxa covered in the study. Making better use of the balanced cohort design and opting for a stratified analysis strategy in favor of pooled analyses is likely to improve the manuscript significantly.

RESPONSE

We appreciate the reviewer for the positive and constructive comments and feedback provided and thank you for the timely review of our manuscript and acknowledgement of the substantial contribution this work provides towards better understanding of gut microbiome composition and function in understudied populations.

We addressed the recommendations and comments as follows, with line numbers noted and the revised sections of the manuscript highlighted.

Suggested major changes

As mentioned above, I think stratifying analyses like the ones presented in Fig. 2-5 by cohort/country would make better use of the study design. I would suggest studying the microbiome associations with SCFA abundance and anthropogenic measures within each cohort first, and then reporting the observed heterogeneity in the observed associations. This is also what is suggested by the current NIH guidelines when studying cohorts that span several ethnic groups (“Reducing Bias and Examining Intervention Effects” at <https://grants.nih.gov/policy/inclusion/women-and-minorities/analyses.htm>). Luckily, the study is in a superb spot to do so due to the balanced cohorts and sufficiently large numbers of individuals in each of the 5 cohorts

RESPONSE:

Thanks for the suggestion of studying the microbiome interactions with SCFA and obesity at the country level to understand country-specific characteristics. We have stratified our analyses and incorporated country-specific analyses for Figures 2-5. We agree that this has improved the manuscript. We summarize our findings below:

Figure 2- In addition to aggregated analysis for country and obese group which was already present in the original manuscript, we have included country specific differential abundance analysis reported as Fig 2c-e and have summarized the results in the results section line 215 to 228 which reads:

“Overall, there was almost no overlap in the features discriminating obese from non-obese groups between the country-specific differential abundance analyses, except for a single *Parabacteroides* ASV that was differentially enriched in non-obese participants in both the Ghanaian and Jamaican cohorts (Figs. 2c, e; ANCOM-BC; $q < 0.05$). When comparing obese and non-obese groups in Ghana, 4 features, *Colidextribacter*, *Butyricococcaceae*, *Oscillospiraceae* and *Parabacteroides* were enriched in the non-obese group (Fig. 2c; ANCOM-BC; $q < 0.05$). The gut microbiota of the obese group in the South African cohort was enriched for 7 ASVs including *Lactobacillus*, *Oribacterium* and *Megasphaera* (Fig. 2d; ANCOM-BC; $q < 0.05$). In the Jamaican non-obese group 6 ASVs were enriched including Christensenellaceae, *Desulfovibrio*, *Eubacterium* and *Parabacteroides*, whereas the relative proportion of *Ruminococcus* was greater in the obese Jamaican group (Fig. 2e; ANCOM-BC; $q < 0.05$). The US non-obese group were enriched for *Intestinimonas* and *Ruminiclostridium* when compared with their obese counterparts (Fig. 2f; ANCOM-BC; $q < 0.05$), whereas there were no significantly enriched features that discriminated obese from non-obese group in participants from Seychelles.”

Figure 3: In addition to aggregated analysis of the microbiomes' prediction accuracy for metabolic features and anthropometrics present in the original manuscript, we have included the same analysis at the country level. The additional results are found in Supplementary Tables 6 – 11. A summary of the results (below) can be found in line 249 to 254 in the results section.

“Similarly, the predictive capacity of gut microbiota features in stratifying individuals by sex or metabolic phenotypes were assessed separately for each of the five study sites. The predictive performance of the model calculated by AUC analysis showed a reduction in accuracy for all parameters determined for all sites (Figs. 3g-k and Supplementary Tables 6 – 11). Obese state was predicted with an AUC = 0.57 for Ghana, AUC = 0.52 for South Africa, AUC= 0.48 for Jamaica, AUC = 0.43 for Seychelles, and the US with an AUC = 0.52 (Figs. 3g-k).”

Figure 4: In addition to aggregated correlation analysis between alpha diversity and individual SCFA and total SCFA, we have included the same analysis within each country and the results summarized below, can be found in line 354 to 363 of the results section. The accompanying figures are in Figure 4b and supplementary Table 14

“When stratified by country, Shannon diversity was negatively associated with acetate concentrations in Jamaica (Supplementary Table 14; Spearman $r = -0.21$, $p < 0.001$) and Seychelles (Supplementary Table 14; Spearman $r = -0.11$; $p = 0.032$), and negatively associated with propionate concentrations in Ghana (Supplementary Table 14; Spearman $r = -0.14$, $p = 0.013$) and South Africa (Supplementary Table 14; Spearman $r = -0.16$, $p = 0.003$). A negative relationship was observed between levels of total SCFA and Shannon in South Africa (Supplementary Table 14; Spearman $r = -0.11$, $p = 0.046$) and Jamaica (Supplementary Table 14; Spearman $r = -0.12$, $p = 0.029$), whereas butyrate correlated with Shannon in South Africa only (Supplementary Table 14; Spearman $r = -0.14$, $p = 0.008$). Finally, valerate levels showed significant correlations with Shannon in all countries (Fig. 4b; Spearman $r = 0.12 - 0.41$, $p < 0.05$).”

Figure 5: In addition to aggregated analysis of the association between microbial features and SCFA, the same analysis has been replicated at each study site. The updated figures are in Figure 5c-g and also summarized in text at line 409 to 429 in the results section which we state below:

“At the country level, several genera contributed to variations in SCFAs. The genera that correlated with SCFA levels for obese and non-obese state differed at each site. For instance, acetate levels correlated negatively with the relative proportions of *Anaerostipes* among obese participants from Ghana and Seychelles and non-obese individuals from Jamaica (Fig. 5c-e; $q < 0.05$). By contrast, positive associations were observed between acetate and *Cantebacterium* among the non-obese US cohort and negative associations with *Coproccoccus* within the non-obese Seychelles group (Fig. 5f, g). Butyrate levels positively correlated with 2 different ASVs assigned to the genus *Subdoligranulum* in all countries except the US; ASV 6915 was positively associated with non-obese individuals in Ghana, South Africa, and Jamaica, whereas ASV 7064 was positively associated with obese individuals in Jamaica and Seychelles, which could be indicative of two different species or functional niche differentiation within a taxon. Similarly, 3 *Blautia* ASVs positively correlated with butyrate levels in non-obese participants from Ghana (ASVs 12508, 12561), South Africa (ASVs 12508, 12561) and Jamaica (ASV 12630). Additionally, ASVs 12822 and 12561 positively correlated with butyrate levels in Jamaican participants irrespective of their obesity status (Fig. 5c-f). Propionate was found to be positively associated with *Prevotella* in the non-obese group from Jamaica and Ghana, while positively correlated with *Blautia* (ASVs 12822, 12561) and *Coproccoccus* (11293) among obese participants from Seychelles, and the US (Fig. 5c-f; $q < 0.05$). In the Ghanaian cohort, valerate positively correlated with *Blautia*, while being inversely associated with *Streptococcus* in the obese group in Jamaica and Seychelles, and with all South African participants (Fig. 5c-e; $q < 0.05$). Collectively, more ASVs correlated with total SCFA among non-obese participants when compared to the obese counterparts.”

Supplementary Fig 3: We have also included country level analysis for the functional pathways with the updated text below found in line 281 to 285.

“On the contrary, when stratified by country, no statistically significant predicted functional pathways differentiated non-obese from obese participants within each country except in Jamaica where only a single predicted pathway (PWY7377) involved in adenosyl cobalamin biosynthesis (anaerobic) was differentially enriched in non-obese individuals (Supplementary Fig 3c, ANCOM-BC; $q < 0.05$).”

Supplementary Fig 4: The updated text for within country level analysis is in line 309 to 320.

“Exploring similar analysis at each study site, several predicted genes including enzymes involved in the pyruvate/acetyl CoA pathway (K01907, K00023, K01640), 4-hydroxybutyrate dehydrogenase (K18120) and 4-hydroxybutyryl-CoA dehydratase (K14534) were differentially enriched in non-obese participants from Ghana (Supplementary Fig. 4c; ANCOM-BC; $q < 0.05$). The relative abundance of two genes both encoding pyruvate/acetyl CoA enzymes (K00171, K00169) were greater among the non-obese South African cohort (Supplementary Fig. 4d; ANCOM-BC; $q < 0.05$), while only a single gene encoding 4-hydroxybutyryl-CoA dehydratase (K14534) was found to be differentially abundant in non-obese individuals from Jamaica (Supplementary Fig. 4e; ANCOM-BC; $q < 0.05$). No statistically significant differences in the proportion of genes encoding enzymes in the butyrate synthesis pathway were observed among participants in the non-obese group compared with the obese counterparts in both Seychelles and the US.”

Supplementary Fig 5: The updated text for within country level analysis is in line 334 to 336.

“When analyzed separately for each country, the relative proportion of predicted genes encoding components of the LPS biosynthesis were not significantly different between non-obese and obese individuals at all 5 study sites.”

In Figure 2, why was the US population chosen as a reference group? I feel that this figure would be more informative by showing the information in panel b) but stratified by country followed by a discussion of overlap and differences between the obesity-associated ASVs.

RESPONSE

We chose the US as reference because US participants consistently present with the worst health outcomes, and it has the highest HDI and highest incidence of obesity amongst the countries included. Therefore, using it as a reference allows us to best understand how different environments result in different health and microbiome outcomes. An explanation has been added to the Figure 2 legend.

We have stratified Figure 2b to country level analyses and the results reported as Fig 2c-e as well as summarized in line 215 to 228 which is detailed above in response to first question.

As the authors observe, gut microbial composition heavily differs across cohorts from different geographic regions, which is particular apparent in 16S studies that are limited to a small highly

variable region of bacterial genomes. Thus, most of the identified ASVs will only be observed in one particular cohort, and only few ASVs will be shared across all cohorts. This creates a block structure in ASVs abundance matrixes that will be readily exploited by any machine learning or regression methods and lead to inflated accuracy metrics as observed by the authors in Fig. 3a. This will also bleed into all other measures that are correlated with the country of origin, such as the outcomes studied in Fig. 3b-f and those confounding effects are hard to get rid of in random forest models. I would expect that the models in Fig. 3b-f would not perform well within each cohort.

In that vein, I found the stratified random forests model in Supplemental Figure 3 the most interesting result in the paper. This was prominently mentioned in the abstract, so it is a bit surprising that it is buried in the supplement. The authors state that microbiome-based prediction performance of anthropogenic measures is related to the epidemiological transition, but I wonder if this might also be confounded by alpha diversity where the RF models have access to more features in some of the cohorts due to the larger number of unique ASVs.

RESPONSE

We thank the reviewer for this perspective and agree that microbial composition varies in the different countries. Indeed, this is supported by our country-specific differential abundance analyses (Fig. 2) where we observe almost no overlap in the features discriminating obese from non-obese groups among the various countries owing to the uniqueness of each population. We have stratified our RF analysis and we observe an overall lower performance of microbiome predictions for metabolic phenotypes at the country level compared to the entire cohort analyses. For instance, in the entire cohort microbiome predicted obesity with an AUC of 0.65 whereas at the country level, obesity state was predicted with an AUC = 0.57 for Ghana, AUC = 0.52 for South Africa, AUC= 0.48 for Jamaica, AUC = 0.43 for Seychelles, and the US with an AUC = 0.52 (Figs. 3g-k) (See above for details).

We moved supplementary Fig. 3 which is a stratified analysis of obese status at the country level to the main text denoted as Fig. 3g-j.

It is an interesting point about the potentially confounding effects of alpha diversity, however we find that the predictive accuracy of each model does not correlate with alpha diversity; for example, while South Africa has significantly greater Shannon diversity than the US, the predictive accuracy for obesity is the same (AUC= 0.52).

Apart from stratifying the analyses in Fig. 5 by country I would also have liked to see how those ASV-SCFA associations relate to the phylogenetic distance of ASVs in some larger taxonomic group, like the bacterial genus. For instance, are bacterial genera associated with SCFA levels across all cohorts more phylogenetically conserved? In case a genus is associated with SCFA production in only one cohort, have the ASVs in this cohort diverged from their respective genus-associated ASVs in the other cohorts? This would allow pinpointing whether there is a signal of potential adaptation in the specific cohorts.

RESPONSE

Thanks for the great suggestion. The results are found in Figure 5c-g and our summary in line 409 to 429 in the main text can be found above in response to the first question.

As we noted in our response above, for example, acetate levels correlated negatively with the relative proportions of an ASV annotated to the genus *Anaerostipes* among obese participants from Ghana and Seychelles and non-obese individuals from Jamaican. In another example from our response above, we observe the same genus (*Subdoligranulum*), but different ASVs correlate positively with butyrate in all countries except US, which we suggest as a functional niche differentiation within the genus.

Since the goal of the study seems to be assessing the relationship between the gut microbiome, SCFA production, and obesity, it is probably missing a figure showing the relationship between SCFA production and obesity stratified by country. As far as I am aware, most of the evidence for lower SCFA abundances/production rates in obese individuals comes from white affluent populations, so it would be interesting whether this relationship is present or absent in the communities studied here. In case the authors are exclusively interested in the relationship between the two through the gut microbiome, mediation analysis would probably have been the better approach.

RESPONSE

We conducted a mediation analysis exploring the association between SCFA and obesity potentially through the microbiome and our results are presented in Supplementary Table 15 and we summarize our findings in line 378 to 386 which reads:

“Based on the biological plausibility of the associations among the gut microbiota, SCFA and obesity (Schwiertz et al. 2010; Dugas, Bernabé, et al. 2018; Riva et al. 2017), and our findings, we applied mediation analysis to evaluate whether the gut microbiota could mediate the relationship between SCFAs and obesity. Our results showed a significant direct effect (ADE) (Supplementary Figure 15; estimate = -0.0003; $p < 2e-16$) and total effect of SCFA (estimate = -0.0003; $p < 2e-16$) (Supplementary Table 15). However, the indirect effect (ACME) of SCFA on obesity through Shannon was not statistically significant ($p > 0.05$) suggesting that the effect of SCFA on obesity cannot be fully explained by the microbiota alpha diversity. When the analysis was stratified by country, the effect of SCFA on obesity diminished (not shown).”

The description of the cohort recruitment could be improved. For instance, it is unclear whether individuals within a country were recruited from a distinct geographic region or were sampled representatively across the general population. It is also unclear how the status of “African origin” was determined. What was the exact definition used here and was that assessment made by the recruiting researcher or the study participants themselves?

RESPONSE

By African origin, we refer to populations that have originated from Africa (ex. African Americans comprised the US cohort) and this status was based on self-report from participants. We have included this information in line 578.

We agree that the manuscript would be greatly improved if the precise sampling locations from the various countries are included. Appropriate edits have been made in text which states in line 581 to 589 as:

“The site in Ghana was based at Nkwantakese, a rural village of approximately 20,000 inhabitants and about 25 km outside of Kumasi. The site in South Africa was in Khayelitsha, an urban informal settlement near Cape Town with over 400,000 inhabitants. The participants from Jamaica were from Spanish Town, an urban area 25 km from the center of Kingston. The Seychelles site was based at Mahé, the largest and most populated of the 100 islands forming the Republic of Seychelles, located approximately 1,500 km east of Kenya in the Indian Ocean and home to approximately 81,000 inhabitants. Participants in the US were recruited in Maywood, IL., a suburb adjacent to the western border of Chicago and home to approximately 24,000 people.”

Suggested minor changes.

Figure 1g does not show the correct data. There are more than 13,000 ASVs, but the figure lists less than 100. So either the figure or the caption is incorrect.

RESPONSE

Thanks so much for identifying this typo and we apologize for that. Legend for Figure 1g has now been rectified and it reads:

“(g) Venn diagram of shared and unique genera between the five countries detected at a relative abundance > 0.001 in more than 50% of the samples.”

Was diet assessed in the study? Because lines 550-553 make it look like it was. If yes, this data should be included in the manuscript and be used to assess whether variation in the microbiome-SCFA/adiposity associations were modulated by dietary patterns.

RESPONSE

Diet was not assessed in this current study. The information in lines 550-553 of the older version of the manuscript has now been removed to avoid confusion. However, we have discussed the absence of dietary information in our manuscript as a study limitation, found in line 555 to 556.

“We had no information on diet and physical activity, lifestyle factors that are well-recognized to have profound influences on the gut microbiota.”

Fig. 4 should also report the correlations for each separate cohort.

RESPONSE

We have added the SCFA correlations with microbiome alpha diversity for each cohort both in Figure 4b and in Supplementary Table 14. The results are summarized in line 354 to 363 of the results section. Details are above.

During the placement in the GreenGenes tree and the PiCrust analysis it should be noted that ASVs from understudied populations are approximated by full genome sequences from overrepresented populations here, which has some inherent bias because the used proxy genomes are unlikely to be a faithful representation of what is actually present in the studied communities (for instance see <https://doi.org/10.1016/j.cell.2021.02.052>). I don't think either adds that much to the manuscript. So I would suggest to either rely on direct measures derived from the ASV sequences at hand or to add a disclaimer to the manuscript that mentions this bias.

RESPONSE

We have followed this recommendation and added a disclaimer that mentions the bias which can be found in line 255 to 260 outlined below:

“The predicted potential microbial functional traits resulting from the compositional differences in microbial taxa between countries and obese state were assessed, although we acknowledge that currently available reference genome databases are likely biased toward well-studied Western populations and may have limited capacity to sufficiently characterize the gut microbiome from understudied populations (Pasolli et al 2019; Tamburini et al 2022).”

As far as I am aware, DEBLUR only uses the forward reads. Was the reported number of reads corrected for that? Why was a paired-end sequencing protocol used even if the reverse reads were omitted in later analyses?

RESPONSE

The paired end protocol is part of our standard pipeline. You are correct that Deblur only uses the forward reads at this time. The reported number of reads refers to the forward reads to reflect what was used in Deblur.

Line 160-162: This should read “a total of 433,364,873 16S V4 amplicon sequences were generated”.

RESPONSE

Thanks so much for the suggestion. We have included “V4” to reflect “a total of 433,364,873 16S rRNA gene V4 region sequences. The amendment is found in line 157.

Line 298: Typo in the reported correlation “(r = 0.0.074)”.

RESPONSE

We have corrected and it now states (SPEARMAN r = 0.074) in line 345

The discussion is a bit excessive relative to the fairly succinct results section, so I would suggest pruning this to a few selected major points.

RESPONSE

We have edited the discussion section to make it more succinct.

Where p-values are reported, that should be accompanied by naming the used test and the effect size.

RESPONSE

We now include the used test and effect size throughout the manuscript.

Strong points of the manuscript

Reviews can be discouraging, especially to early career scientists. Thus, to end the review on a positive note, here is a list of parts where I think the study and manuscript were exceptionally well-crafted. The introduction makes a great job at summarizing a large body of research and pointing out the challenges. The study cohort was excellently assembled, included local institutions and stakeholders, and has added them as authors. Very few studies have that level of awareness for social equity. I love how the authors made sure that all samples were processed in few batches, at the same site and with consistent methodology. Even though fecal SCFAs have been measured before, few studies make the efforts to do so for samples from diverse populations, as this requires transporting volatile samples over large distances or introducing biases due to the different processing sites. The standardized measures here will be of interest to many researchers. I appreciate that all the data from the manuscript was provided and was accessible during review.

RESPONSE

The authors thank the reviewer for the positive appraisal and their kind words about our manuscript, and for highlighting the significant contributions and relevance of studying SCFAs, microbiome composition and function in diverse understudied populations to the field.

Reviewer #2 (Remarks to the Author):

In this manuscript, the authors leverage the METS cohort – a well-phenotyped multi-country study of adults of African origin – to explore associations between the gut microbiome, short-

chain fatty acids (SCFA), and obesity. The authors profiled the microbiome (using 16S sequencing) and levels of SCFA in >1,800 fecal samples – a remarkable effort. The profiling was done according to the best practices in the field. Based on these alone, this is a remarkable resource, and it would be great to see it shared with the scientific community. From a biological perspective, progress in understanding the predictive utility of the microbiome in various pathologies and its role in their manifestation has been hampered by the under-representation of diverse populations. The METS cohort, which profiles individuals from countries representing a broad spectrum of income levels, offers an attractive opportunity to test whether the microbiome (and its metabolites) are universal markers of obesity. The authors do reach a conclusion regarding the predictive ability of the microbiome. However, I would suggest a closer look at the interpretation (see below). I would also like to see the discussion of the role of SCFA expanded further (see below). Beyond these two points, I have some comments regarding the study design that I would like to see the authors address in the text. Overall, a remarkable achievement!

RESPONSE

We thank the reviewer for the positive and constructive comments and feedback provided and we appreciate the timely review of our manuscript and acknowledgement of the substantial contribution this work provides towards better understanding of gut microbiome composition and function in understudied populations.

We addressed the recommendations and comments as follows, with line numbers noted and the revised sections of the manuscript highlighted.

Main comments

A. Microbiome studies focused on non-US, non-Western Europeans are critically needed to understand whether a given disease is associated with a universal microbiome signature (that can be thus harnessed for prediction or can be targeted) or, rather, distinct populations exhibit unique markers for the same disease. From this perspective, this study design is suitable to address this important question in the context of obesity. The authors' conclusion is that "...microbiota differences between obesity and non-obesity may be larger in low-to-middle-income countries compared to high-income countries", and again in the discussion, "the utility of the microbiota in predicting whether an individual was lean or obese was inversely correlated with the income-level of the country of origin." I have summarized in the table below the authors' findings. I am finding it difficult to find support for this claim in the data. Observed ASVs and unweighted beta diversity partly support this claim, but the weighted beta diversity and the area under the receiver-operator curve do not.

Notably, obesity is best predicted by the microbiome by Ghana (low income), followed by the US (highest income) – this does not follow the epidemiological transition. This is such an important take-home message that I think deserves more attention. I agree with the authors regarding a potential underlying cause for the discrepancy between weighted and unweighted

beta diversity, but this does not explain why the AUROC does not follow the epidemiological transition.

The authors should try to identify the factors that allow better prediction in Ghana, South Africa, and the US, but not Jamaica and Seychelles. Even if no such factors are identified, the conclusions of the analyses should be adjusted to better represent the data, throughout the manuscript.

Ghana South Africa Jamaica Seychelles US

Observed ASVs Sig. Sig. NS NS NS

Weighted BDiv NS NS NS NS Sig.

Unweighted BDiv Sig. Sig. NS NS NS

AUROC 0.75 0.61 0.49 0.57 0.64

AUROC Rank 1 3 5 4 2

AUROC Trend

RESPONSE

We thank the reviewer for sharing this great perspective, and we agree that additional factors would have to be identified to support our conclusion that obesity predicted by the microbiome follow an epidemiological transition. After reanalysis, the relationship between income status and prediction accuracy is less clear, while a relationship does exist. We have therefore followed the reviewer's recommendation and have reduced the emphasis on this particular result. We have modified aspects of our abstract, discussion and conclusions to reflect that.

For example, line 43 to 45 of the abstract, we state:

“Country of origin is accurately predicted from the fecal microbiota (AUC=0.97), while the predictive accuracy for obesity differs between countries, being greatest in Ghana (AUC = 0.57).

Also, in line 438 to 440 in the discussion we state:

“the utility of the microbiota in predicting whether an individual was non-obese or obese differed considerably by country of origin, being greatest in Ghana and lowest in Jamaica and the Seychelles.”

B. The manuscript aims to address an important open question in the field of microbiome and metabolic syndrome – whether higher levels of SCFAs are associated with improved or poorer metabolic health (in this case, obesity). The authors find that “When compared to their obese counterparts, non-obese participants had significantly higher weight-adjusted fecal total and individual SCFA levels”. What I’m still missing here is: A) can obesity be predicted according to SCFA levels (total, and each SCFA) – similar to what the authors achieved with the microbiome? B) Are there differences between the countries in the ability of SCFA to predict obesity? This would be such a valuable finding, regardless of the result.

RESPONSE

Thank you for this great question and we agree that the predictive performance of SCFA for obesity is important perspective to include irrespective of the findings. Although we detect a significant decrease in SCFA levels associated with obesity, prediction capacity based on SCFA data (total, individual) is limited and not improved stratifying by country. The prediction of obesity by SCFA within the entire cohort and in each country is poor and both results are comparable. We have reported the results in Supplementary Fig. 6 and summarized in text in line 364 to 377 found below:

“Using Xgboost machine learning model, the predictive capacity of SCFAs to stratify individuals to either obese or non-obese in the entire cohort was assessed. The predictive performance of the model was calculated by area under the receiver operating characteristic curve (AUC) analysis, which showed poor accuracy and similar outcomes for the different SCFAs. Total SCFA predicted obese state with AUC = 0.55, acetate and propionate with AUC = 0.53, butyrate with AUC = 0.52 and valerate with AUC = 0.51 (Supplementary Fig. 6). Similar analysis to determine obesity status was performed at the country-specific level and the results of the predictive performance of the model were comparable to that of the entire cohort analysis (Supplementary Fig. 6). The comparative predictive capacity for obese state was higher in Ghana (AUC = 0.60) using valerate; higher in south Africa (AUC = 0.55) using propionate; higher in Jamaica using acetate (AUC = 0.56) and total SCFA (AUC = 0.56). Obese state predicted by butyrate was comparable among all countries (AUC = 0.51) except Ghana (AUC = 0.46). Overall, the predictive capacity of SCFAs were higher in Ghana, South Africa and Jamaica compared with the US and Seychelles (Supplementary Fig. 6).”

C. Study design. Unclear if antibiotic treatment was an exclusion criterion, I could not find any indication of that. Please address that directly in the methods. If antibiotics use was not an exclusion criterion, please address in your discussion how this might have impacted the results, or preferably, validate the results after omitting antibiotics-associated samples, if data for this purpose are available.

RESPONSE

Antibiotic usage was an exclusion criterion and text has been updated in line 605.

D. Study design. Unclear how stool samples were kept before transport to the research clinic (home refrigerator, freezer?), how were they transferred (coolers? Room temperature?), what was the maximum allowed age of sample (assuming not all patients had a sample the day of the visit – how many days of storage were allowed?). The age of the sample, and the temperature in which it was stored and transported can significantly impact the microbiome, metabolites, volatiles, and SCFA. Please add relevant information to the methods. Please address in the discussion how this might have impacted the results.

RESPONSE

We have updated the text in lines 626 to 629.

“Participants were encouraged to provide stool samples in clinic or just prior to clinic visits using a standard collection kit (EasySampler stool collection kit, Alpco, NH). In cases where this was not possible, participants stored stool samples in home freezers or coolers for 1-3 days prior to clinic visits.”

Additional comments

1. *Abstract, “lowest and highest end of the epidemiologic transition spectrum, respectively” – It might be worthwhile to clarify what the “epidemiological transition spectrum” means or use alternative, more common terms. It’s defined in the introduction but complicates interpretation of the abstract.*

RESPONSE

We have modified the phrase to use more common terms. The sentence in line 39 to 41 now reads:

“Fecal microbiota diversity and SCFAs are greatest in Ghanaians, and lowest in the US population, representing the lowest and highest end of the urbanization spectrum, respectively.”

2. *Introduction, “A major driver of obesity is the adoption of a western lifestyle, which is characterized by excessive consumption of ultra-processed foods.” – please add a reference here.*

RESPONSE

We have expanded on the western lifestyle characteristics and included references and it now reads in the text in lines 53 to 56:

“A major driver of obesity is the adoption of a western lifestyle, which is characterized by excessive consumption of ultra-processed foods combined with low level of physical activity and increased sedentary time (Ecklu-Mensah 2022; Abdelaal, le Roux, and Docherty 2017; Ajala et al. 2017).”

3. *Introduction, regarding the conflicting evidence on SCFA in obesity, it might be worthwhile to mention Perry, Nature 2016; Tirosh, Sci Transl Med 2019.*

RESPONSE

We have included these references in line 99 to 100:

(Schwartz et al. 2010; Rahat-Rozenbloom et al. 2014; Teixeira et al. 2013, Perry et al 2016; Tirosh et al 2019).”

4. Introduction, Line 50-51 – I advise adjusting the language here when discussing deaths related to obesity. It is not simply ‘obesity’ that accounts for 60% of deaths related to high body mass index, as obesity is describing a high body mass index. Rather, it is comorbidities associated with obesity that are accounting for these deaths.

RESPONSE

We thank you for the suggestion. We have modified the text, which can be found in lines 50 to 53 and below:

“Obesity remains an ongoing global health epidemic that continues to worsen at an alarming rate, affecting more than 600 million adults worldwide (“Obesity and Overweight” n.d.), including over a third of Americans (Hales et al. 2020). Importantly, comorbidities associated with obesity account for over 60% of deaths worldwide (Tseng and Wu 2019).”

5. Methods, “HIV-positive individuals”, please use person-first language.

RESPONSE

We have amended the text which can be found in line 603 to 606 to read:

“Participants were excluded from participating in the original METS study if they self-reported being persons with an infectious disease including HIV, being pregnant, breast-feeding, using antibiotics within 3 months or having any condition which prevented the individual from participating in normal physical activities.”

6. Methods, Please include registration numbers for the trials.

RESPONSE

The registration number has been included (NCT03378765) and can be found in line 603.

7. Methods, please consider including the STORMS microbiome checklist: <https://www.stormsmicrobiome.org/>

RESPONSE

We have included the STORMS microbiome checklist.

8. Methods, “METS data showed Ghanaians consumed the greatest amount...”, why is this section here?

RESPONSE

We apologize for the confusion and have subsequently removed the text from the methods section since diet was not assessed in the current study. However, we have discussed the absence of dietary information in our manuscript as a study limitation, found in line 555 to 556.

“We had no information on diet and physical activity, lifestyle choices that are well-recognized to have profound influences on the gut microbiota.”

9. *Methods, lines 498-500, this sentence is unclear.*

RESPONSE

We appreciate the reviewer pointing out this area of confusion. We have modified the text which can be found in line 595 to 597 as follows:

“This framework has allowed us to understand how increasing global Westernization, resulting in a greater consumption of ultra-processed foods, is associated with higher prevalence of obesity, type 2 diabetes and cardiometabolic diseases.”

10. *When citing panels from figures in text, please denote which panel in the figure is being discussed.*

RESPONSE

We have modified accordingly.

11. *Bacterial family names should be italicized.*

RESPONSE

We have italicized all Family, Genus and Species names.

12. *Figure 3 - for ease of understanding, please title each graph with the variable that is being modeled. Additionally, is there any significance to the difference between macro- and micro-average in some of the ROC curves? For example, diabetes status (Fig. 3C) seems to be well explained by the macro-average as compared to the micro-average. Also, for these models, a confusion matrix might be a good supplemental figure to provide for classifier metrics and additional understanding to the readers.*

RESPONSE

We have substantially modified Figure 3 to reflect parameters used in the model. We have also included supplementary Tables 4, 5, 7-11 for the confusion matrix tables for entire cohort and country specific analyses. These have been indicated in text at lines 235-239, 252.

Micro-averaging values are impacted by data imbalance since it averages across each sample whereas Macro-averaging provides equal weight to the characterization of each sample. We report macro-averaging values in the manuscript. We have included this statement in the legend for Fig. 3.

13. ANCOM-BC is referred to as 'ANCOM-BC' and 'ANCOMBC' in the text; this should be 'ANCOM-BC' and consistent throughout.

RESPONSE

The manuscript has been updated to reflect ANCOM-BC throughout.

14. Supplemental Figures 5 and 6 - it would be worth having the pathway name in addition to or replacement of the KO pathway number.

RESPONSE

We have added the pathway names and are now designated as Supplementary Figs. 3 and 4

REVIEWER COMMENTS

Reviewer #1 (Remarks to the Author):

The authors have addressed all the previously mentioned concerns and recommendations and implemented them into the revised manuscript. Consequently, the manuscript has now been improved significantly and is in a pretty good shape. There are some odd results in the Random Forest models that would need to be addressed, but I am confident the authors will be able to do so.

Line 230 states that “Microbial taxonomic features predict obesity overall and within each country” but the second part of that statement is not supported by Figure 3 or the Supplementary Tables 6-11. A model randomly assigning labels would have an AUC of 0.5 and all the country-specific models have AUCs close to that value. So my conclusion would be that microbial taxonomic features do **not** predict obesity-related measures within each country. The Supplementary Tables 5-11 also seem to imply that the fitted random forest models simply assign all individuals into a single group independent of the data (for instance, see the non-diabetic group in the confusion matrix in the SI tables 5 and 6). This would be a prime example of a failed model. So the text should be adjusted to reflect the negative results that have been observed here.

Line 709 states that the mediation analysis was run with “R (v4.5.0)” which is not an existing version of R. The most recent version of R is 4.3.1. So I think this is a typo.

Reviewer #2 (Remarks to the Author):

The revised manuscript addresses all my comments on the original submission, and I thank the authors for the additional analyses and text revisions. Based on the new analyses, I request that the authors address two follow-up points.

1. When each country is analyzed individually, the ability to predict obesity based on the microbiome or SCFA is not much better than a coinflip in some cases. This begs the question of whether this is unique to the relationship between microbiome features/metabolites and obesity or, rather, a general loss of predictive power (e.g., due to a lower N compared to the whole cohort). It would be

helpful to include the AUC values for microbiome-based predictions of the other variables (sex, diabetes, glucose, hypertension) per country. If the predictive ability remains robust (compared with the whole cohort), this should be highlighted in the discussion - that the loss of predictive power is specific to obesity and microbiome/SCFA. Rather, if the prediction is notably worse for any parameter, this should also be included in the discussion as one should be cautious not to over-interpret the per-country analyses.

2. Despite a significant inverse correlation between obesity and SCFA levels, the ability to predict obesity based on SCFA, even with data from the entire cohort, is not much better than chance. I find this intriguing and noteworthy; As the authors correctly state in their introduction, there is an ever-growing interest in the role of SCFA in metabolic health, and they are a major focus of this study. I would like to see the authors place more emphasis on the finding that SCFA do not predict obesity, what might be the underlying causes, and the potential significance to debate on SCFA role in metabolic health.

NCOMMS-23-15272B. Gut microbiota and fecal short chain fatty acids differ with adiposity and country of origin: The METS-Microbiome Study

REVIEWER COMMENTS

Reviewer #1 (Remarks to the Author):

The authors have addressed all the previously mentioned concerns and recommendations and implemented them into the revised manuscript. Consequently, the manuscript has now been improved significantly and is in a pretty good shape. There are some odd results in the Random Forest models that would need to be addressed, but I am confident the authors will be able to do so.

*Line 230 states that “Microbial taxonomic features predict obesity (-related measures) overall and (but not within each country) within each country (change title, leave country..)” but the second part of that statement is not supported by Figure 3 or the Supplementary Tables 6-11. A model randomly assigning labels would have an AUC of 0.5 and all the country-specific models have AUCs close to that value. So my conclusion would be that microbial taxonomic features do *not* predict obesity-related measures within each country. The Supplementary Tables 5-11 also seem to imply that the fitted random forest models simply assign all individuals into a single group independent of the data (for instance, see the non-diabetic group in the confusion matrix in the SI tables 5 and 6). This would be a prime example of a failed model. So the text should be adjusted to reflect the negative results that have been observed here.*

Line 709 states that the mediation analysis was run with “R (v4.5.0)” which is not an existing version of R. The most recent version of R is 4.3.1. So I think this is a typo.

Reviewer #2 (Remarks to the Author):

The revised manuscript addresses all my comments on the original submission, and I thank the authors for the additional analyses and text revisions. Based on the new analyses, I request that the authors address two follow-up points.

1. When each country is analyzed individually, the ability to predict obesity based on the microbiome or SCFA is not much better than a coinflip in some cases. This begs the question of whether this is unique to the relationship between microbiome features/metabolites and obesity or, rather, a general loss of predictive power (e.g., due to a lower N compared to the whole cohort). It would be helpful to include the AUC values for microbiome-based predictions of the other variables (sex, diabetes, glucose, hypertension) per country. If the predictive ability remains robust (compared with the whole cohort), this should be highlighted in the discussion - that the loss of predictive power is specific to obesity and microbiome/SCFA. Rather, if the prediction is notably worse for any parameter, this should also be included in the discussion as one should be cautious not to over-interpret the per-country analyses.

2. Despite a significant inverse correlation between obesity and SCFA levels, the ability to predict obesity based on SCFA, even with data from the entire cohort, is not much better than chance. I find this intriguing and noteworthy; As the authors correctly state in their introduction, there is an ever-growing interest in the role of SCFA in metabolic health, and they are a major focus of this study. I would like to see the authors place more emphasis on the finding that SCFA do not predict obesity, what might be the underlying causes, and the potential significance to debate on SCFA role in metabolic health.

Point by point response to reviewers comments

Reviewer #1 (Remarks to the Author):

The authors have addressed all the previously mentioned concerns and recommendations and implemented them into the revised manuscript. Consequently, the manuscript has now been improved significantly and is in a pretty good shape. There are some odd results in the Random Forest models that would need to be addressed, but I am confident the authors will be able to do so.

Line 230 states that “Microbial taxonomic features predict obesity (-related measures) overall and (but not within each country) within each country (change title, leave country..)” but the second part of that statement is not supported by Figure 3 or the Supplementary Tables 6-11. A model randomly assigning labels would have an AUC of 0.5 and all the country-specific models have AUCs close to that value. So my conclusion would be that microbial taxonomic features do **not** predict obesity-related measures within each country. The Supplementary Tables 5-11 also seem to imply that the fitted random forest models simply assign all individuals into a single group independent of the data (for instance, see the non-diabetic group in the confusion matrix in the SI tables 5 and 6). This would be a prime example of a failed model. So the text should be adjusted to reflect the negative results that have been observed here.

RESPONSE: We agree that reporting negative results is equally important. We have edited the text to reflect that only in Ghana was the predictive accuracy marginally better than chance. All other countries show no predictive accuracy. This has been changed in the abstract, results and discussion.

For example, our subheading in line 231 now reflects “*Microbial taxonomic features predict obesity overall but not within each country*”

Also, in line 253 to 259 we state: “For example, obese state was marginally predictive only for Ghana (AUC=0.57) while all other countries lost accuracy (Figs. 3g-k). The predictive accuracy (Supplementary Table 6) for diabetes status was only retained for Ghana (AUC=0.69) and Jamaica (AUC=0.66); glucose status prediction was lost for all countries but South Africa, where it improved (AUC 0.78); and prediction of hypertension was only retained for Ghana (AUC=0.63). The predictive ability for sex was maintained for all countries (Supplementary Table 6).”

Line 709 states that the mediation analysis was run with “R (v4.5.0)” which is not an existing version of R. The most recent version of R is 4.3.1. So I think this is a typo.

RESPONSE: We were referring to the mediation package version which is v4.5.0 – to avoid confusion we have edited this line to read: “The mediation package (v4.5.0) in R (v4.3.0) was used to infer causal relationships between gut microbial diversity, SCFAs and obesity”

Reviewer #2 (Remarks to the Author):

The revised manuscript addresses all my comments on the original submission, and I thank the authors for the additional analyses and text revisions. Based on the new analyses, I request that the authors address two follow-up points.

1. When each country is analyzed individually, the ability to predict obesity based on the microbiome or SCFA is not much better than a coinflip in some cases. This begs the question of whether this is unique to the relationship between microbiome features/metabolites and obesity or, rather, a general loss of predictive power (e.g., due to a lower N compared to the whole cohort). It would be helpful to include the AUC values for microbiome-based predictions of the other variables (sex, diabetes, glucose, hypertension) per country. If the predictive ability remains robust (compared with the whole cohort), this should be highlighted in the discussion - that the loss of predictive power is specific to obesity and microbiome/SCFA. Rather, if the prediction is notably worse for any parameter, this should also be included in the discussion as one should be cautious not to over-interpret the per-country analyses.

RESPONSE: We did provide that information in supplementary table 6 (below). We have now made this clear in the text. We also included discussion of Table S6 in the text – highlighting, as the reviewer suggested some interesting findings:

Supplementary Table 6. The classification accuracy of gut microbiota for estimating metabolic disease indicators using a Random Forest model at each study site.

	Diabetes Status	Glucose Status	HTN Status	Sex
Ghana	0.69	0.55	0.63	0.76
South Africa	0.57	0.78	0.54	0.74
Jamaica	0.66	0.58	0.56	0.63
Seychelles	0.54	0.56	0.48	0.75
USA	0.53	0.59	0.47	0.66

Line 250 to 259 reads: “Similarly, the predictive capacity of gut microbiota features in stratifying individuals by sex or metabolic phenotypes were assessed separately for each of the five study sites. The predictive performance of the model calculated by AUC analysis showed changes in accuracy for all parameters determined for all sites (Figs. 3g-k and Supplementary Tables 6 – 11). For example, obese state was marginally predictive only for Ghana (AUC=0.57) while all other countries lost accuracy (Figs. 3g-k). The predictive accuracy (Supplementary Table 6) for diabetes status was only retained for Ghana (AUC=0.69) and Jamaica (AUC=0.66); glucose status prediction was lost for all countries but South Africa, where it improved (AUC 0.78); and prediction of hypertension was only retained for Ghana (AUC=0.63). The predictive ability for sex was maintained for all countries (Supplementary Table 6).”

We also added a section to the discussion (440-449):

“Importantly, the utility of the microbiota in predicting whether an individual was non-obese or obese differed considerably by country of origin, being marginally better than chance only in Ghana and not predictive for all other countries. Interestingly, only sex was universally predicted at individual sites; while predictive accuracy for diabetes status was only retained for Ghana (AUC=0.69) and Jamaica (AUC=0.66); glucose status only in South Africa (AUC 0.78); and hypertension was only retained for Ghana (AUC=0.63), suggesting that predicting metabolic disease indicators from the microbiome was impacted by differences in this relationship between countries. Importantly, fecal SCFA concentrations could not predict obesity either globally or within

each, which suggests that the relationship with SCFA and obesity is still unclear, and SCFA may be a poor biomarker for obesity.”

We modified the abstract (line 42 to 46) to read: “Diabetes, glucose state, hypertension, obesity, and sex can be accurately predicted from the global microbiota, but when analyzed at the level of country, predictive accuracy is only universally maintained for sex. Diabetes, glucose, and hypertension are only predictive in certain low-income countries. Our findings suggest that adiposity-related microbiota differences differ between low-to-middle-income compared to high-income countries.”

2. Despite a significant inverse correlation between obesity and SCFA levels, the ability to predict obesity based on SCFA, even with data from the entire cohort, is not much better than chance. I find this intriguing and noteworthy; As the authors correctly state in their introduction, there is an ever-growing interest in the role of SCFA in metabolic health, and they are a major focus of this study. I would like to see the authors place more emphasis on the finding that SCFA do not predict obesity, what might be the underlying causes, and the potential significance to debate on SCFA role in metabolic health.

RESPONSE: We agree – while we have not added this to the abstract (due to the word limit), we have added a discussion on this to the discussion.

For example, in line 447 to 449 we state: “Importantly, fecal SCFA concentration could not predict obesity either globally or within each site, which suggests that the relationship with SCFA and obesity is still unclear and SCFA may be a poor biomarker for obesity.”

Also line 529 to 540 states “Our study, due to the size and diversity of the cohort, provides robust evidence to suggest that fecal SCFA concentrations are not predictive of obesity status and that fecal SCFA may function as a poor biomarker for obesity. Previous studies have suggested that measures of both circulating and fecal SCFAs could be more reliable prognostic markers of obesity status (Wang et al. 2020; Müller et al. 2019; de la Cuesta-Zuluaga, Mueller, et al. 2018; Sanna et al. 2019), a hypothesis that remain to be fully elucidated in our study cohort. Additionally, controlled human intervention studies, including the quantitation of whole-body turnover rates of SCFAs (Perry et al 2016) are needed to ascertain the potential health benefits before clinical translation can be implemented to improve metabolic health. Broadly, our findings prompt caution in relying on fecal microbial metabolites alone to infer obesity outcomes, since obesity is a heterogenous construct with several unique mechanisms involving host-related factors such as genetic predisposition and microbial SCFAs involved in precipitating disease susceptibility.”

Line 591 to 593 reads: “While SCFA concentrations could not predict obese status, obese individuals had different gut microbiota composition and function compared to non-obese individuals.”